# AlterMOMA: Fusion Redundancy Pruning for Camera-LiDAR Fusion Models with Alternative Modality Masking

**Shiqi Sun**[*1], **Yantao Lu**[†*1], **Ning Liu**[*2], **Bo Jiang**[3], **Jinchao Chen**[1], **Ying Zhang**[1]

[1] Department of Computer Science, Northwestern Polytechnical University
[2] Midea Group
[3] Didi Chuxing

{shiqisun, yantaolu, cjc, ying_zhang}@nwpu.edu.cn
ningliu1220@gmail.com
boj.horizon@gmail.com

## Abstract

Camera-LiDAR fusion models significantly enhance perception performance in autonomous driving. The fusion mechanism leverages the strengths of each modality while minimizing their weaknesses. Moreover, in practice, camera-LiDAR fusion models utilize pre-trained backbones for efficient training. However, we argue that directly loading single-modal pre-trained camera and LiDAR backbones into camera-LiDAR fusion models introduces similar feature redundancy across modalities due to the nature of the fusion mechanism. Unfortunately, existing pruning methods are developed explicitly for single-modal models, and thus, they struggle to effectively identify these specific redundant parameters in camera-LiDAR fusion models. In this paper, to address the issue above on camera-LiDAR fusion models, we propose a novelty pruning framework **Alter**native **Mo**dality **Ma**sking Pruning (AlterMOMA), which employs alternative masking on each modality and identifies the redundant parameters. Specifically, when one modality parameters are masked (deactivated), the absence of features from the masked backbone compels the model to *reactivate* previous redundant features of the other modality backbone. Therefore, these redundant features and relevant redundant parameters can be identified via the reactivation process. The redundant parameters can be pruned by our proposed importance score evaluation function, **Alter**native **Eva**luation (AlterEva), which is based on the observation of the loss changes when certain modality parameters are activated and deactivated. Extensive experiments on the nuScenes and KITTI datasets encompassing diverse tasks, baseline models, and pruning algorithms showcase that AlterMOMA outperforms existing pruning methods, attaining state-of-the-art performance.

## 1 Introduction

Camera-LiDAR fusion models are prevalent in autonomous driving, effectively leveraging the sensor properties, including the accurate geometric data from LiDAR point clouds and the rich semantic context from camera images [1, 2], providing a more comprehensive understanding of the environment [3, 4]. However, the exponential increase in parameter counts due to fusion architectures introduces significant computational costs, especially when deploying these systems on resource-

---

[*]Joint first authorship. Either author can be cited first.
[†]Corresponding author.

38th Conference on Neural Information Processing Systems (NeurIPS 2024).

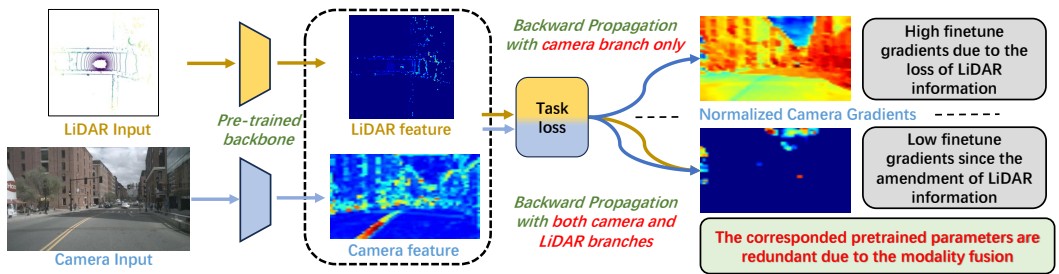

Figure 1: **Motivating example** of fusion-redundant features in the 3D object detection task. We employ backward propagation on camera-LiDAR fusion models with pre-trained backbones to observe the gradient difference (features utilization) between with camera backbone only and with both the camera and LiDAR backbone. Notably, certain pre-trained parameters in the camera backbone are redundant due to the amendment of LiDAR information. It reveals that similar feature extraction exists across modalities, which introduces additional redundancy when camera-LiDAR fusion models directly loads single-modal pre-trained backbones.

constrained edge devices, which is a crucial challenge for autonomous driving [5]. Network pruning is one of the most attractive methods for addressing the challenge above of identifying and eliminating redundancy in models. Existing pruning algorithms target single-modal models [6, 7, 8, 9, 10, 11, 12] or multi-modal models that merge distinct types of data [13, 14], such as visual and language inputs. However, it's important to note that directly applying these algorithms to camera-LiDAR fusion models can lead to significant performance degradation. The degradation can be reasoned for two main factors that existing pruning methods overlooked: 1) the key fusion mechanism specific to vision sensor inputs within models, and 2) the training scheme where models typically load single-modal pre-trained parameters onto each backbone [2, 15]. Specifically, since single-modality models lack the cross-modality fusion mechanism, existing pruning algorithms traditionally do not consider inter-modality interactions. Furthermore, because the pre-trained backbones (image or LiDAR) are trained separately, they are not optimized jointly, exacerbating the redundancy in features extracted from each backbone. Though leveraging pre-trained backbone improves the training efficiency compared with models training from scratch, we argue that *directly loading single-modal pre-trained camera and LiDAR backbones into camera-LiDAR fusion models introduces similar feature redundancy across modalities due to the nature of the fusion mechanism.*

In detail, since backbones are independently pre-trained on single-modal datasets, they extract features comprehensively, which leads to similar feature extraction across modalities. Meanwhile, the fusion mechanism selectively leverages reliable features while minimizing weaker ones across modalities to enhance model performance. This selective utilization upon similar feature extraction across modalities introduces the additional redundancy: *Each backbone independently extracts similar features, which subsequent fusion modules will not potentially utilize.* For instance, both camera and LiDAR backbones extract geometric features to predict depth during pre-training. However, geometric features extracted from the LiDAR backbone are considered more reliable during fusion because LiDAR input data contain more accurate geometric information than the cameras, e.g., object distance, due to the physical properties of sensors. Consequently, this leads to the redundancy of geometric features of the camera backbone. In summary, similar feature extraction across modalities, coupled with the following selective utilization in fusion modules, leads to two counterparts of similar features across modalities: those utilized by fusion modules in one modality (i.e., fusion-contributed), and those that are redundant in the other modality (i.e., fusion-redundant). We also illustrate the fusion-redundant features in Figure 1.

To address the above challenge, we propose a novel pruning framework **AlterMOMA**, specifically designed for camera-LiDAR fusion models to identify and prune fusion-redundant parameters. AlterMOMA employs alternative masking on each modality, followed by observing loss changes when certain modality parameters are activated and deactivated. These observations serve as important indications to identify fusion-redundant parameters, which are integral to our importance scores evaluation function, **AlterEva**. Specifically, the camera and LiDAR backbones are alternatively masked. During this process, the absence of fusion-contributed features and relevant parameters in the masked (deactivated) backbone compels the fusion modules to reactivate their fusion-redundant

counterparts from the other backbone. Throughout this reactivation, changes in loss are observed as indicators for contributed and fusion-redundant parameters across modalities. These indicators are then combined in AlterEva to maximize the importance scores of contributed parameters while minimizing the scores of fusion-redundant parameters. Then, parameters with low importance scores will be pruned to reduce computational costs.

To validate the effectiveness of our proposed framework, extensive experiments are conducted on several popular 3D perception datasets with camera and LiDAR sensor data, including nuScenes [16] and KITTI [17]. These datasets encompass a range of 3D autonomous driving tasks, including 3D object detection, tracking, and segmentation.

The contributions of this paper are as follows: 1) We propose a pruning framework **AlterMOMA** to effectively compress camera-LiDAR fusion models 2) we propose an importance score evaluation function **AlterEva**, which identifies fusion-redundant features and their relevant parameters across modalities 3) we validate the effectiveness of the proposed AlterMOMA on **nuScenes** and **KITTI** for 3D detection and segmentation tasks.

## 2 Related Work

**Camera-LiDAR Fusion.** With the advancement of autonomous driving technology, the efficient fusion of diverse sensors, particularly cameras and LiDARs, has become crucial [18, 19]. Fusion architectures can be categorized into three types based on the stage of fusion within the learning framework: early fusion [20, 21], late fusion [22], and intermediate fusion [2, 3, 15]. Current state-of-the-art (SOTA) fusion models evolve primarily within intermediate fusion and combine low-level machine-learned features from each modality to yield unified detection results, thus significantly enhancing perception performance compared with early or late fusion. Specifically, camera-LiDAR fusion models focus on aligning the camera and LiDAR features through dimension projection at various levels, including point [23], voxel [24], and proposal [3]. Notably, the SOTA fusion paradigm aligns all data to the bird's eye view (BEV) [2, 4, 15, 25, 26], has gained traction as an effective approach to maximize the utilization of heterogeneous data types.

**Network Pruning.** Network pruning effectively compresses deep models by reducing redundant parameters and decreasing computational demands. Pruning algorithms have been well-explored for single-modal perception tasks [27, 28, 29, 30, 31, 32], focusing on evaluating importance scores to identify and remove redundant parameters or channels. These scores are based on data attributes [7, 33], weight norms [34, 35], or feature map ranks [28]. However, single-modal pruning algorithms are not suited for the complexities of camera-LiDAR fusion models. While some multi-modal pruning algorithms exist [13, 14], they are mainly designed for models combining different data types like language and vision. Therefore, there is a pressing need for pruning algorithms specifically devised for camera-LiDAR fusion models. From the perspective of granularity, pruning algorithms can be divided into two primary categories: 1) structured pruning, which entails removing entire channels or rows from parameter matrices, and 2) unstructured pruning, which focuses on eliminating individual parameters. For practical applications, we have adapted our method to support both types of pruning.

## 3 Methodology

### 3.1 Preliminaries

We firstly review some basic concepts including camera-LiDAR fusion models and pruning formulation. Camera-LiDAR fusion models consist of 1) a LiDAR feature extractor $\mathbf{F}_l$ to extract features from point cloud inputs, 2) a camera feature extractor $\mathbf{F}_c$ to extract features from image inputs, 3) the fusion module and following task heads $\mathbf{F}_f$ to get the final task results. The parameters denote as $\theta = \{ \theta_l, \theta_c, \theta_f \}$ for LiDAR backbone, camera backbone, and fusion and task heads, respectively. Take camera backbone for instance, $\theta_c = \{\theta_c^1, \theta_c^2, ..., \theta_c^{N_c}\}$ denotes all weights in the camera backbone, where $N_c$ represents the total number of parameters in camera backbone. Therefore, for the LiDAR input $\mathbf{X}_l$ and camera input $\mathbf{X}_c$, the training process of models could be denoted as

$$\arg \min_{\theta_{l,c,f}} \mathcal{L}(\mathbf{Y}, \mathbf{F}_f(\theta_f; \mathbf{F}_l(\theta_l; \mathbf{X}_l), \mathbf{F}_c(\theta_c; \mathbf{X}_c)), \tag{1}$$

where $\mathbf{Y}$ denotes the ground truth, and $\mathcal{L}$ represents the task-specific loss functions.

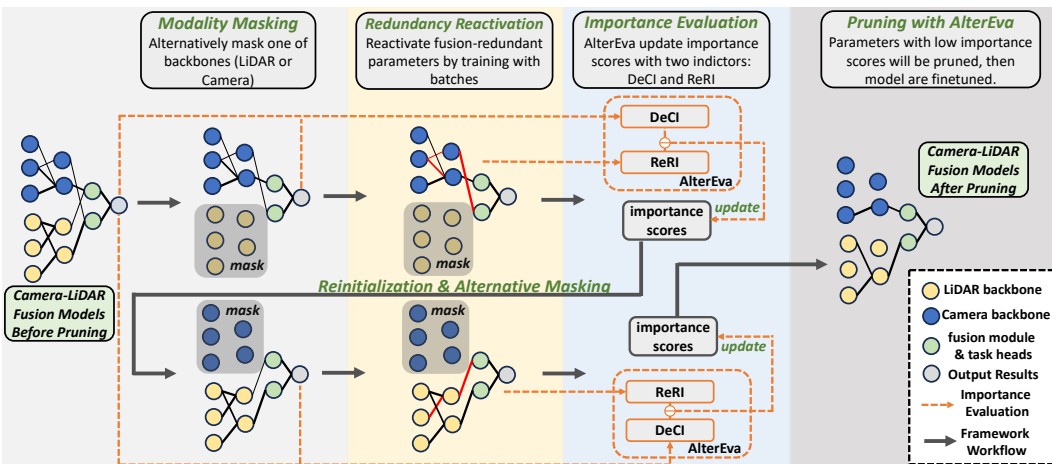

Figure 2: **Overview of the AlterMOMA** : The framework begins with *Modality Masking*, where one of the backbones is initially masked. This step is followed by *Redundancy Reactivation* and *Importance Evaluation*, where the parameter importance scores are initially calculated with AlterEva. Afterward, the models undergo *Reinitialization* and *Alternative Masking* of the other backbone, leading to another round of *Redundancy Reactivation* and *Importance Evaluation*. When scores of all parameters in backbones are calculated fully with AlterEva (detailed in Section 3.3), models are pruned to remove parameters with low importance scores and then finetuned. Notably, we use *black* lines to represent parameters of models and *red* lines to represent reactivated fusion-redundant parameters. The thickness of these lines indicates the contribution of parameters.

Importance-based pruning typically involves using metrics to evaluate the importance scores of parameters or channels. Subsequently, optimization methods are employed to prune the parameters with lower importance scores, that are nonessential within the model. For the camera-LiDAR fusion models, the optimization process can be formulated as follows:

$$\arg\max_{\delta_{ij}} \sum_{i\in\{l,c,f\}} \sum_{j=1}^{N_i} \delta_{ij} \mathbf{S}\big(\theta_i^j\big), \; s.t. \sum_{i\in\{l,c,f\}} \sum_{j=1}^{N_i} \delta_{ij} = k, \tag{2}$$

where $\delta_{ij}$ is an indicator which is 1 if $\theta_i^j$ will be kept or 0 if $\theta_i^j$ is to be pruned. $\mathbf{S}$ is designed to measure the importance scores for parameters, and $k$ represents the kept parameter number, where $k = (1 - \rho) \cdot \sum_{i\in\{l,c,f\}} N_i$ with the pruning ratio $\rho$.

### 3.2 Overview of Alternative Modality Masking Pruning

Similar feature extraction across modalities, coupled with the selective utilization of features in the following fusion modules introduce redundancy in camera-LiDAR fusion models. Therefore, similar features and their relevant parameters can be categorized into two counterparts across modalities: those that contribute to fusion and subsequent task heads (fusion-contributed), and those that are redundant (fusion-redundant). In this section, we propose the pruning framework AlterMOMA, which alternatively employs masking on camera and Lidar backbones to identify and remove the fusion-redundant parameters. AlterMOMA is developed based on a novel insight: *"The absence of fusion-contributed features will compel fusion modules to 'reactivate' their fusion-redundant counterparts as supplementary, which, though less effective, are necessary to maintain functionality."* For instance, if the LiDAR backbone is masked, the previously fusion-contributed geometric features it provided are absent. To fulfill the need for accurate position predictions, the model still needs to process geometric features. Consequently, the fusion module is compelled to utilize the geometric features from the unmasked camera backbone, which were previously fusion-redundant. We refer to this process as *Redundancy Reactivation*. By observing changes during this *Redundancy Reactivation*, fusion-redundant parameters can be identified. The overview of AlterMOMA is shown in Figure 2, and the detailed steps are in Algorithm 1 of Appendix D. The key steps are introduced as follows:

**Modality Masking.** Three binary masks are denoted as $\mu_l$, $\mu_c$, and $\mu_f \in \{0, 1\}$, correspond to the parameters applied separately on the LiDAR backbone, the camera backbone, and the fusion and

tasks head. Our framework begins by masking either one of the camera backbones or the LiDAR backbone. Here we take masking the LiDAR backbone as the illustration. The masks are with $\mu_l = 0$, $\mu_c = 1$, $\mu_f = 1$. The camera backbone will be masked alternatively.

**Redundancy Reactivation.** To allow masked models to reactivate fusion-redundant parameters, we train masked models with batches of data. Specifically, $B$ batches of data $\mathcal{D}_i$, $i \in \{1, 2, ..., B\}$ are sampled from the multi-modal dataset $\mathcal{D}$.

**Importance Evaluation.** After *Redundancy Reactivation*, the importance scores of parameters in the camera backbones are calculated with our proposed importance score evaluation function AlterEva detailed in Section 3.3. Since fusion modules need to consider the reactivation of both modalities, the importance scores of parameters in the fusion module and task heads will be updated once the importance scores of both the camera and Lidar backbones' parameters are calculated.

**Alternative Masking.** After *Importance Evaluation* of camera modality, models will reload the initialized parameters, and then the other backbone will alternatively be masked, with $\mu_l = 1$, $\mu_c = 0$, $\mu_f = 1$. Then the step *Redundancy Reactivation* and *Importance Evaluation* will be processed again to update the importance scores of parameters in the LiDAR backbone and the fusion module.

**Pruning with AlterEva.** After evaluating the importance scores using AlterEva, parameters with low importance scores are pruned with a global threshold determined by the pruning ratio. Once the pruning is finished, the model is fine-tuned with the task-specific loss, as indicated in Eqn. 1.

## 3.3 Alternative Evaluation

In this section, we will detail the formulation of our proposed AlterEva, which consists of two distinct indicators to evaluate the parameter importance scores. As outlined in section 3.2, the importance scores are alternatively calculated with AlterEva in *Importance Evaluation*. Then, parameters with low importance scores are removed in the pruning process. The goal of AlterEva is to maximize the scores of parameters that contribute to task performance while minimizing the scores of fusion-redundant parameters. To achieve this, AlterEva incorporates two key indicators: 1) **Deactivated Contribution Indicator** (DeCI) evaluate the parameter contribution to the overall task performance of the fusion models, 2) **Reactivated Redundancy Indicator** (ReRI) identifies fusion-redundant parameters across both modalities. Since changes in loss can directly reflect the parameter contribution difference to task performance during alternative masking, both indicators are designed based on the observation of loss decrease or increase, when certain modality parameters are activated or deactivated. Specifically, take parameters in the camera backbone as an instance, DeCI observe the loss increases with masking camera backbone itself, while ReRI observe loss decrease with masking LiDAR backbone and reactivating camera backbone via *Redundancy Reactivation*. Formally, we formulate the loss for the fusion models with masks. With three binary masks and the dataset defined in Section 3.2, the loss is denoted as follows, by simplifying some of the extra notations used in Eqn. 1:

$$\mathcal{L}_m(\mu_c, \mu_l, \mu_f; \mathcal{D}) = \mathcal{L}(\mu_l \odot \theta_l, \mu_c \odot \theta_c, \mu_f \odot \theta_f; \mathcal{D}). \tag{3}$$

For brevity, we assume $\mu_c = 1$, $\mu_l = 1$, and $\mu_f = 1$, and we only specify in the formulation when a mask is zero. For example, $\mathcal{L}_m(\mu_c = 0; \mathcal{D})$ indicates that $\mu_c = 0$, $\mu_f = 1$ and $\mu_l = 1$. Since the alternative masking is performed on both backbones, we illustrate our formulation by calculating two indicators for parameters in the camera backbone.

**Deactivated Contribution Indicator.** If a parameter is important and contributes to task performance, deactivating this parameter will lead to task performance degradation, which will be reflected in an increase in loss. Therefore, to derive the contribution of the $i$-th parameter $\theta_c^i$ of the camera backbone, we observe the changes in loss when this parameter is deactivating via masking, denoted as follows:

$$\hat{\Phi}_{\theta_c^i} = |\mathcal{L}_m(; \mathcal{D}) - \mathcal{L}_m(\mu_c^i = 0; \mathcal{D})|, \tag{4}$$

where $\mu_c^i$ represents the mask for $\theta_c^i$, and $\hat{\Phi}_{\theta_c^i}$ denotes the indicator DeCI for $\theta_c^i$. However, the total number of parameters is enormous, deactivating and evaluating each parameter independently are computationally intractable. Therefore, we design an alternative efficient method to approximate the evaluation in Eqn. 4 by leveraging the Taylor first-order expansion inspired by [36]. We first observe the loss changes $|\mathcal{L}_m(; \mathcal{D}) - \mathcal{L}_m(\mu_c = 0; \mathcal{D})|$ by deactivating the entire camera backbones. Then, the first-order approximation of evaluation in Eqn. 4 is calculated by expanding the loss change in each individual parameter $\theta_c^i$ with Taylor expansion, considering $\theta_c = \{\theta_c^1, ..., \theta_c^{N_c}\}$. This method

allows us to estimate the contribution for each parameter, denoted as follows:

$$\hat{\Phi}_{\theta_c^i} = \left| \mathcal{L}_m(;\mathcal{D}) + \mu_c^i \odot \theta_c^i \cdot \frac{\partial \mathcal{L}_m(;\mathcal{D})}{\partial \theta_c^i} - \mathcal{L}_m(\mu_c = 0;\mathcal{D}) - \mu_c^i \odot \theta_c^i \cdot \frac{\partial \mathcal{L}_m(\mu_c = 0;\mathcal{D})}{\partial \theta_c^i} \right|. \quad (5)$$

When $\theta_c$ is deactivating with $\mu_c = 0$, $\mu_c^i = 0$ for $i \in \{1, ..., N^c\}$, which means that the last term of Eqn. 5 is zero. Meanwhile, when considering importance scores on a global scale, the $\mathcal{L}_m(;\mathcal{D})$ and $\mathcal{L}_m(\mu_c = 0;\mathcal{D})$ can be treated as constant for all $\theta_c^i$. Thus the first term and the third term can be disregarded. Therefore, the final indicator of each parameter's contribution, represented by our DeCI, can be expressed as follows:

$$\hat{\Phi}_{\theta_c^i} = \left| \theta_c^i \cdot \frac{\partial \mathcal{L}_m(;\mathcal{D})}{\partial \theta_c^i} \right|. \quad (6)$$

This formulation enables tractable and efficient computation without *Modality Masking* of the camera backbone itself, achieved by performing a single backward propagation in the *Importance Evaluation* with initialized parameters.

**Reactivated Redundancy Indicator.** As discussed in Section 3.2, the identification of fusion-redundant parameters relies on our understanding of the fusion mechanism: when fusion-contributed features from the LiDAR backbone are absent due to masking, the previously fusion-redundant counterparts and their relevant parameters from the camera backbone will be reactivated during the *Redundancy Reactivation*. Therefore, to reactivate and identify fusion-redundant parameters in the camera backbone, the *Modality Masking* of the LiDAR backbone ($\mu_l = 0$) and *Redundancy Reactivation* are processed first. Throughout this process, the loss evolves from $\mathcal{L}_m(\mu_l = 0;\mathcal{D}_1)$ to $\mathcal{L}_m(\mu_l = 0;\mathcal{D}_B)$, and the parameters evolve from $\theta_{c,0}$ (i.e. $\theta_c$) to $\theta_{c,B}$. Similar to the formulation of DeCI, we observe the decrease in loss during *Redundancy Reactivation* and refer to this observation as our ReRI, denoted as follows:

$$\tilde{\Phi}_{\theta_c} = |\mathcal{L}_m(\mu_l = 0;\mathcal{D}) - \mathcal{L}_m(\mu_l = 0;\mathcal{D}_1) + ... + \mathcal{L}_m(;\mathcal{D}_{B-1}) - \mathcal{L}_m(\mu_l = 0;\mathcal{D}_B)|$$
$$= |\mathcal{L}_m(\mu_l = 0;\mathcal{D}) - \mathcal{L}_m(\mu_l = 0;\mathcal{D}_B)|. \quad (7)$$

Specifically, this process is designed to identify parameters that contribute to the task performance of models with the masked LiDAR backbone, highlighting those that are fusion-redundant. Since we want to observe reactivation rather than parameters updating of this masked model across training batches, we apply the first-order Taylor expansion to the initial $i$-th parameters $\theta_{c,0}^i$, denoted as:

$$\tilde{\Phi}_{\theta_{c,0}^i} = \left| \mathcal{L}_m(\mu_l = 0;\mathcal{D}) + \mu_c^i \odot \theta_{c,0}^i \cdot \frac{\partial \mathcal{L}_m(\mu_l = 0;\mathcal{D})}{\partial \theta_{c,0}^i} \right.$$
$$\left. - \mathcal{L}_m(\mu_l = 0;\mathcal{D}_B) - \mu_c^i \odot \theta_{c,0}^i \cdot \frac{\partial \mathcal{L}_m(\mu_l = 0;\mathcal{D}_B)}{\partial \theta_{c,0}^i} \right|. \quad (8)$$

To derive the gradient on initial parameters $\theta_{c,0}^i$ of the last term, we could use the chain rule and write out based on the gradient of the last step,

$$\frac{\partial \mathcal{L}_m(\mu_l = 0;\mathcal{D}_B)}{\partial \theta_{c,0}^i} \cdot \theta_{c,0}^i = \frac{\partial \mathcal{L}_m(\mu_l = 0;\mathcal{D}_B)}{\partial \theta_{c,B}^i} \prod_{j=1}^{B} \frac{\partial \theta_{c,j}^i}{\partial \theta_{c,j-1}^i} \cdot \theta_{c,0}^i \approx \frac{\partial \mathcal{L}_m(\mu_l = 0, \mathcal{D}_B)}{\partial \theta_{c,B}^i} \cdot \theta_{c,0}^i. \quad (9)$$

According to the Proposition A in the Appendix A, this approximation is reached by dropping some small terms with sufficiently small learning rates [9]. Since $\theta_c^i$ is activating with $\mu_c^i = 1$, and the $\mathcal{L}_m(\mu_l = 0;\mathcal{D})$ and $\mathcal{L}_m(\mu_c = 0;\mathcal{D}_B)$ can be treated as constant for all $\theta_c^i$, we could denote our final formulation by simplifying Eqn. 8, and denoted as follow:

$$\tilde{\Phi}_{\theta_c^i} = \left| \theta_c^i \cdot \frac{\partial \mathcal{L}_m(\mu_l = 0;\mathcal{D})}{\partial \theta_c^i} - \theta_c^i \cdot \frac{\partial \mathcal{L}_m(\mu_l = 0;\mathcal{D}_B)}{\partial \theta_{c,B}^i} \right|, \quad (10)$$

where $\theta_c^i$ is the $\theta_{c,0}^i$ and $\tilde{\Phi}_{\theta_c^i}$ represent the ReRI for $\theta_c^i$.

To the goal of parameters with significant contributions maintaining high importance scores while those identified as fusion-redundant are assigned lower scores, AlterEva calculate the final importance scores by subtracting ReRI from DeCI. Since DeCI of parameters in the camera backbone could be calculated without masking the camera backbone itself, DeCI and ReRI of camera parameters could be calculated in the same alternative masking stage (with LiDAR backbone masking), which simplifies the process of our framework AlterMOMA. Therefore, with the combination Eqn. 6 and Eqn. 10, the AlterEva of the camera backbones could be presented with the normalization:

$$S(\theta_c^i) = \alpha \cdot \frac{\hat{\Phi}_{\theta_c^i}}{\sum_{j=0}^{N_c} \hat{\Phi}_{\theta_c^j}} - \beta \cdot \frac{\tilde{\Phi}_{\theta_c^i}}{\sum_{j=0}^{N_c} \tilde{\Phi}_{\theta_c^j}}, \quad (11)$$

where $\alpha$ and $\beta$ are the hyper parameters and $\mathcal{S}(\theta_c^i)$ represent the importance score evaluation function AlterEva for $\theta_c^i$. Similarly, the AlterEva of parameters in the LiDAR backbones (i.e. $\theta_l$) and in the fusion modules (i.e. $\theta_f$) could be derived as:

$$\mathcal{S}(\theta_l^i) = \alpha \cdot \frac{\hat{\Phi}_{\theta_l^i}}{\sum_{j=0}^{N_l} \hat{\Phi}_{\theta_l^j}} - \beta \cdot \frac{\tilde{\Phi}_{\theta_l^i}}{\sum_{j=0}^{N_l} \tilde{\Phi}_{\theta_l^j}}, \tag{12}$$

$$\mathcal{S}(\theta_f^i) = \alpha \cdot \frac{\hat{\Phi}_{\theta_f^i}}{\sum_{j=0}^{N_f} \hat{\Phi}_{\theta_f^j}} - \frac{\beta}{2} \cdot \frac{\tilde{\Phi}_{\theta_f^i}(\mu_l = 0)}{\sum_{j=0}^{N_f} \tilde{\Phi}_{\theta_f^j}(\mu_l = 0)} - \frac{\beta}{2} \cdot \frac{\tilde{\Phi}_{\theta_f^i}(\mu_c = 0)}{\sum_{j=0}^{N_f} \tilde{\Phi}_{\theta_f^j}(\mu_c = 0)}, \tag{13}$$

where $\tilde{\Phi}_{\theta_l^i}$ and $\tilde{\Phi}_{\theta_f^i}(\mu_c = 0)$ is calculated when camera backbone is masking, while $\tilde{\Phi}_{\theta_f^i}(\mu_l = 0)$ is calculated with LiDAR backbone masking. AlterEva can efficiently calculate importance scores with backward propagation, enhancing the tractability of AlterMOMA. For brevity, we omit the derivations related to parameters in the LiDAR backbone and fusion modules, but additional details are available in Appendix B and Appendix C.

## 4 Experimental Results

### 4.1 Baseline Models and Datasets

To validate the efficacy of our proposed framework, empirical evaluations were conducted on several camera-LiDAR fusion models, including the two-stage detection models AVOD-FPN [3], as well as the end-to-end architecture based on BEV space, such as BEVfusion-mit [15] and BEVfusion-pku [2]. For AVOD-FPN, the point cloud input is processed using a voxel grid representation, while all input views are extracted using a modified VGG-16 [37]. Notably, the experiment on the AVOD-FPN demonstrates the efficiency of AlterMOMA on two-stage models, although this isn't the SOTA fusion architecture for recent 3D perception tasks. Current camera-LiDAR fusion models are moving towards a unified architecture that extracts camera and LiDAR features within a BEV space. Thus, our primary results focus on BEV-based unified architectures, specifically BEVfusion-mit [15] and BEVfusion-pku [2]. We conducted tests using various backbones. For camera backbones, we included Swin-Transformer (Swin-T) [38] and ResNet [39]. For LiDAR backbones, we used SECOND [18], VoxelNet [40] and PointPillars [41].

We perform our experiments for both 3D object detection and BEV segmentation tasks on the KITTI [17] and nuScenes [16], which are challenging large-scale outdoor datasets devised for autonomous driving tasks. The KITTI dataset contains 14,999 samples in total, including 7,481 training samples and 7,518 testing samples, with a comprehensive total of 80,256 annotated objects. To adhere to standard practice, we split the training samples into a training set and a validation set in approximately a 1:1 ratio and followed the difficulty classifications proposed by KITTI, involving *easy*, *medium*, and *hard*. NuScenes is characterized by its comprehensive annotation scenes, encompassing tasks including 3D object detection, tracking, and BEV map segmentation. Within this dataset, each of the annotated 40,157 samples presents an assemblage of six monocular camera images, adept at capturing a panoramic 360-degree field of view. This dataset is further enriched with the inclusion of a 32-beam LiDAR scan, amplifying its utility and enabling multifaceted data-driven investigations.

### 4.2 Implementation Details

We conducted the 3D object detection and segmentation experiments with MMdetection3D [42] on NVIDIA RTX 3090 GPUs. To ensure fair comparisons, consistent configurations of hyperparameters were employed across different experimental groups. To train the 3D object detection baselines, we utilize Adam as the optimizer with a learning rate of 1e-4. We employ Cosine Annealing as the parameter scheduler and set the batch size to 2. For BEV segmentation tasks, we employ Adam as the optimizer with a learning rate of 1e-4. We utilize the one-cycle learning rate scheduler and set the batch size to 2. The hyperparameters $\alpha$ and $\beta$ in Section 3.3 are both set with 1. The baseline pruning methods include IMP [43], SynFlow [44], SNIP [35], and ProsPr [9], with the hyperparameters specified in their papers respectively.

Table 1: **3D object detection performance comparison with the state-of-the-art pruning methods** on the nuScenes validation dataset. We list the mAP and NDS of models pruned by different approaches within 80%, 85%, and 90% pruning ratios. The two baseline models are trained with SwinT and VoxelNet backbone.

| Baseline Model | BEVfusion-mit | | | | | | BEVfusion-pku | | | | | |
|---|---|---|---|---|---|---|---|---|---|---|---|---|
| Sparsity | 80% | | 85% | | 90% | | 80% | | 85% | | 90% | |
| Metric | mAP | NDS | mAP | NDS | mAP | NDS | mAP | NDS | mAP | NDS | mAP | NDS |
| [No Pruning] | 67.8 | 70.7 | - | - | - | - | 66.9 | 70.4 | - | - | - | - |
| IMP | 59.3 | 66.8 | 51.2 | 59.2 | 42.7 | 50.4 | 57.3 | 65.9 | 49.8 | 57.7 | 40.7 | 47.2 |
| SynFlow | 63.2 | 67.9 | 56.9 | 64.1 | 49.3 | 58.7 | 62.4 | 67.1 | 55.4 | 63.2 | 47.6 | 57.1 |
| SNIP | 62.2 | 67.5 | 56.4 | 63.6 | 50.2 | 58.8 | 61.8 | 67.5 | 54.7 | 62.9 | 47.8 | 57.3 |
| ProsPr | 64.3 | 69.6 | 61.9 | 66.1 | 58.6 | 62.5 | 63.6 | 68.4 | 59.9 | 66.1 | 56.7 | 62.7 |
| **AlterMOMA (Ours)** | **67.3** | **70.2** | **65.5** | **69.5** | **63.5** | **66.7** | **66.5** | **70.1** | **64.2** | **68.1** | **62.3** | **66.0** |

Table 2: **3D object detection performance comparison with the state-of-the-art pruning methods** on the KITTI Validation dataset on car class. We list the models pruned by different approaches within 80%, and 90% pruning ratios. The baseline model is AVOD-FPN architecture.

| Sparsity | 80% | | | | | | 90% | | | | | |
|---|---|---|---|---|---|---|---|---|---|---|---|---|
| Task | AP-3D | | | AP-BEV | | | AP-3D | | | AP-BEV | | |
| Difficulty | Easy | Moderate | Hard | Easy | Moderate | Hard | Easy | Moderate | Hard | Easy | Moderate | Hard |
| Car [No Pruning] | 82.4 | 72.2 | 66.5 | 89.4 | 83.9 | 78.7 | - | - | - | - | - | - |
| IMP | 65.8 | 57.7 | 51.3 | 69.2 | 64.6 | 59.7 | 52.1 | 45.7 | 43.2 | 59.6 | 54.2 | 51.7 |
| SynFlow | 74.2 | 65.7 | 60.2 | 79.5 | 75.3 | 70.3 | 64.5 | 54.7 | 48.1 | 73.5 | 67.6 | 64.4 |
| SNIP | 73.5 | 64.9 | 59.8 | 79.1 | 75.8 | 69.6 | 62.7 | 52.3 | 45.8 | 72.4 | 66.9 | 63.7 |
| ProsPr | 78.9 | 69.6 | 62.1 | 85.2 | 79.1 | 75.7 | 74.2 | 63.4 | 59.1 | 81.2 | 75.1 | 71.9 |
| **AlterMOMA (Ours)** | **80.5** | **70.2** | **63.2** | **87.2** | **81.5** | **77.9** | **77.4** | **68.2** | **62.3** | **85.3** | **79.9** | **75.8** |

## 4.3 Experimental Results on Unstructured Pruning

To evaluate the efficiency of AlterMOMA with unstructured pruning, we conduct experiments across multiple fusion architectures and datasets for 3D object detection and semantic segmentation. Specifically, to evaluate the efficiency of AlterMOMA on two-stage fusion architectures, we applied AlterMOMA to AVOD-FPN [3], using the KITTI dataset with the task of 3D detection. Besides, BEVfusion-mit [15] and BEVfusion-pku [2], as two representative camera-LiDAR fusion models with unified BEV-based architectures, are applied with AlterMOMA using the nuScenes dataset to validate the efficiency on both 3D detection and semantic segmentation tasks. Additionally, to validate the robustness of AlterMOMA with various backbones, we conducted experiments with alternative images and point backbone, including ResNet [39] and PointPillars [41].

**3D Object detection on nuScenes with BEV-based fusion Architectures.** The experimental results are presented in Table 1. Note that baseline models are BEVfusion-mit trained with SwinT and VoxelNet backbone. As reported in Table 1, single-modal pruning methods, including IMP, SynFlow, SNIP, and ProsPr, experience significant declines in accuracy performance. Even the ProsPr, considered the best-performing method among these single-modal pruning techniques, demonstrates the mAP decrease of 3.5% in accuracy at the 80% pruning ratio and 9.2% at the 90% pruning ratio on BEVfusion-mit. Conversely, the incorporation of our AlterMOMA yielded promising results. For example, comparing with the baseline pruning method ProsPr, AlterMOMA boosts the mAP of BEVfusion-mit by 3.0% (64.3 → 67.3), 3.6% (61.9 → 65.5), and 4.9% (58.6 → 63.5) for the three different pruning ratios. Similarly, AlterMOMA obtains much higher mAP and NDS than the other four pruning baselines with different pruning ratios on BEVFusion-mit and BEVFusion-pku.

**3D Object detection on KITTI with the two-stage fusion architecture.** To validate the efficiency of AlterMOMA on the two-stage detection fusion architecture, we conduct experiments with various pruning ratios on KITTI with AVOD-FPN architecture as the baseline. The experimental results are presented in Table 2. Specifically, Table 2 presents the results for the car class on the KITTI, detailing AP-3D and AP-BEV across various difficulty levels including *easy*, *moderate*, and *hard*. Existing pruning methods experience significant declines in performance on different metrics of different difficulties. Even the best-performing method among single-modal pruning methods, ProPr, shows a decrease in AP-3D of 3.5%, 2.6%, and 4.4% in the *easy*, *moderate*, and *hard* difficulty levels, respectively, at the 80% pruning ratio. Conversely, the AlterMOMA has yielded

Table 3: **BEV segmentation performance comparison** on the nuScenes validation dataset.

| Sparsity | 80% mIoU | 85% mIoU | 90% mIoU |
|---|---|---|---|
| **[No Pruning]** | 61.8 | - | - |
| IMP | 53.2 | 51.8 | 49.9 |
| SynFlow | 56.5 | 55.3 | 53.1 |
| SNIP | 55.9 | 54.9 | 53.2 |
| ProsPr | 57.7 | 56.2 | 54.1 |
| **AlterMOMA** | **60.7** | **59.2** | **57.7** |

Table 4: **3D object detection performance with various backbones** on the nuScenes validation dataset.

| Sparsity | 80% mAP | 85% mAP | 90% mAP |
|---|---|---|---|
| [No Pruning] | 53.7 | - | - |
| IMP | 47.1 | 43.3 | 37.8 |
| SynFlow | 49.5 | 45.8 | 40.3 |
| SNIP | 49.7 | 45.5 | 41.2 |
| ProsPr | 50.1 | 47.5 | 44.1 |
| **AlterMOMA** | **51.7** | **50.6** | **48.3** |

Table 5: **3D object detection performance of structure pruning** on the nuScenes validation dataset.

| Sparsity | ResNet101 + SECOND | | |
|---|---|---|---|
| | mAP | NDS | GFLOPs($\downarrow$ %) |
| **[No Pruning]** | 64.6 | 69.4 | 610.66 |
| IMP-30% | 60.8 | 67.2 | 428.7 (29.8) |
| ProsPr-30% | 64.2 | 69.1 | 413.4 (32.3) |
| **AlterMOMA-30%** | **65.3** | **69.9** | **420.13 (31.2)** |
| IMP-50% | 57.6 | 65.2 | 297.39 (51.3) |
| ProsPr-50% | 62.5 | 68.4 | 285.79 (53.2) |
| **AlterMOMA-50%** | **64.5** | **69.5** | **264.42 (56.7)** |

promising results. For instance, comparing with the pruning method ProsPr, AlterMOMA enhances both the AP-3D and AP-BEV on AVOP-FPN by 3.2% (74.2% $\rightarrow$ 77.4%) and 3.9% (81.2% $\rightarrow$ 85.3%) for *easy* difficulties at the 90% pruning ratio. Furthermore, AlterMOMA consistently outperforms the other four pruning baselines across various difficulties and pruning ratios on AVOD-FPN. The comprehensive experimental results on 3D object detection validate the effectiveness of our AlterMOMA across different camera-LiDAR fusion architectures.

**3D Semantic Segmentation on nuScenes** To validate the robustness of our work, we extend our performance evaluation of AlterMOMA to the semantic-centric BEV map segmentation task. Note that baseline models are BEVfusion-mit trained with SwinT and VoxelNet backbone. We use the nuScenes dataset and utilize the BEV map segmentation validation set. The pivotal evaluation metric for this task is the mean Intersection over Union (mIoU). with the experimental configuration detailed in the work by [15], we perform our evaluation on the BEVfusion-mit, as shown in Table 3. We observed that existing pruning methods still meet a significant accuracy drop by 8.6% (IMP), 5.3% (SynFlow), 5.9% (SNIP), and 4.1% (ProsPr) for the 80% pruning ratio. Alternatively, our proposed approach yields significant advancements in performance. Specifically, compared with the ProsPr, AlterMOMA achieves a substantial enhancement by 3.0% (57.7% $\rightarrow$ 60.7%), 3.0% (56.2% $\rightarrow$ 59.2%), and 3.6% (54.1% $\rightarrow$ 57.7%) for the three different pruning ratios. These results empirically prove the efficacy of our pruning algorithms when applied to the BEV map segmentation task.

**Results on Various Backbone Architectures** To comprehensively assess the efficacy of AlterMOMA, we conducted experiments with alternative images and point backbones which will influence fusion. Specifically, we replaced the original VoxelNet backbone with PointPillar and the SwinT backbone with ResNet101 on the architecture of BEVFusion-mit. As depicted in Table 4, the results obtained from these experiments consistently demonstrate state-of-the-art performance with various pruning ratios. Particularly noteworthy is the achievement of substantial 1.6%, 3.1%, and 4.2% improvement compared to the ProsPr baseline under the pruning ratio of 80%, 85% and 90%. This consistent improvement is observed across different pruning ratios, affirming the effectiveness of AlterMOMA with different backbones employed. These outcomes robustly demonstrate the general applicability of AlterMOMA to various backbone architectures.

### 4.4 Structure Pruning Results on 3D object Detection

To assess the efficacy of our proposed pruning approach in structure pruning, we conducted experiments with BEVfusion-mit models with ResNet101 as the camera backbone and SECOND as the LiDAR backbones. Specifically, We measured the performance of the pruned networks with a similar amount of FLOP reductions and reported the number of FLOPs('GFLOPs'). As depicted in Table 5, the results obtained from these experiments consistently demonstrate state-of-the-art performance with various pruning sparsities. Our evaluations at 30% and 50% pruning sparsities reveal that Alter-MOMA not only maintains a competitive mAP and NDS but also achieves a substantial reduction in computational overhead. Notably, with the 30% pruning sparsities, AlterMOMA achieves a 0.7% on mAP and 0.5% on NDS compared with the unpruned baseline models, which reveals that removing similar feature redundancy improves the efficiency of models. Specifically, compared with the ProsPr baseline, AlterMOMA achieves a substantial enhancement by 1.1% ($64.6\% \rightarrow 65.3\%$), and 2.0% ($62.5\% \rightarrow 64.5\%$), for the two different pruning sparsities.

## 5 Discussion and Conclusion

Although our approach identifies similar feature redundancy in camera-LiDAR fusion models, it is limited to the perception field. Extending it to other multi-modal models, such as vision-language models, requires further research. Fusion modules across various modalities exhibit different functionalities. In multi-sensor fusion models (camera, LiDAR, and Radar), the focus is on supplementing and spatially aligning data by leveraging the sensors' physical properties, fusing low-level features. However, in models with disparate data types like vision and language, fusion modules focus on matching high-level semantic contexts. Therefore, AlterMOMA primarily addresses redundancy from supplementary functionality in multi-sensor fusion perception architectures.

In this paper, we explore the computation reduction of camera-LiDAR fusion models. A pruning framework AlterMOMA is introduced to address redundancy in these models. AlterMOMA employs alternative masking on each modality and observes loss changes when certain modality parameters are activated and deactivated. These observations are integral to our importance scores evaluation function AlterEva. Through extensive evaluation, our proposed framework AlterMOMA achieves better performance, surpassing the baselines established by single-modal pruning methods.

## Acknowledgments

This work was supported in part by the National Natural Science Foundation of China (62106202, 62102316 62403386), and in part by the Key Research and Development Projects of Shaanxi Province (2024GX-YBXM-118, 2024GX-YBXM-254), and the Aeronautical Science Foundation of China (2023M073053003).

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

# A Derivation of Eqn.9 in Section 3.3

**Proposition.** *For a camera-LiDAR fusion model with parameters $\theta_c$ for the camera backbone and parameters $\theta_l$ for the LiDAR backbone, we can mask one of the backbones using masks $\mu_l = 0$ for the LiDAR backbone and $\mu_c = 0$ for the camera backbone. Take the models with masking LiDAR backbone as instance. With a sufficiently small learning rate $\epsilon$, the masked model is trained with batches of data $\mathcal{D}_i$, $i \in \{1, 2, ..., B\}$ sampled from the dataset $\mathcal{D}$. We assume the parameters update from $\theta_{c,0}$ to $\theta_{c,B}$, and the loss changes from $\mathcal{L}_m(\mu_l = 0)$ to $\mathcal{L}_m(\mu_l = 0; \mathcal{D}_B)$. Then we could get the equation denoted as:*

$$\frac{\partial \mathcal{L}_m(\mu_l = 0; \mathcal{D}_B)}{\partial \theta_{c,0}^i} \cdot \theta_{c,0}^i \approx \frac{\partial \mathcal{L}_m(\mu_l = 0, \mathcal{D}_B)}{\partial \theta_{c,B}^i} \cdot \theta_{c,0}^i. \tag{14}$$

*Proof.* We can extend the left side of the equation using the chain rule, denoted as,

$$\frac{\partial \mathcal{L}_m(\mu_l = 0; \mathcal{D}_B)}{\partial \theta_{c,0}^i} \cdot \theta_{c,0}^i = \frac{\partial \mathcal{L}_m(\mu_l = 0; \mathcal{D}_B)}{\partial \theta_{c,B}^i} \cdot \frac{\partial \theta_{c,B}^i}{\partial \theta_{c,0}^i} \cdot \theta_{c,0}^i$$

$$= \frac{\partial \mathcal{L}_m(\mu_l = 0; \mathcal{D}_B)}{\partial \theta_{c,B}^i} \cdot \frac{\partial \theta_{c,B}^i}{\partial \theta_{c,B-1}^i} \cdot .... \cdot \frac{\partial \theta_{c,2}^i}{\partial \theta_{c,1}^i} \cdot \frac{\partial \theta_{c,1}^i}{\partial \theta_{c,0}^i} \cdot \theta_{c,0}^i \tag{15}$$

$$= \frac{\partial \mathcal{L}_m(\mu_l = 0; \mathcal{D}_B)}{\partial \theta_{c,B}^i} \cdot \left[ \prod_{j=1}^{B} \frac{\partial \theta_{c,j}^i}{\partial \theta_{c,j-1}^i} \right] \cdot \theta_{c,0}^i.$$

Due to the updates with the learning rate $\epsilon$, we can represent $\theta_{c,j}^i$ as follows:

$$\theta_{c,j}^i = \theta_{c,j-1}^i - \epsilon \cdot \frac{\partial \mathcal{L}_m(\mu_l = 0; \mathcal{D}_{j-1})}{\partial \theta_{c,j-1}^i}. \tag{16}$$

Combining with Eqn. 15 and Eqn. 16, we could derive as following:

$$\frac{\partial \mathcal{L}_m(\mu_l = 0; \mathcal{D}_B)}{\partial \theta_{c,0}^i} \cdot \theta_{c,0}^i = \frac{\partial \mathcal{L}_m(\mu_l = 0; \mathcal{D}_B)}{\partial \theta_{c,B}^i} \cdot \left[ \prod_{j=1}^{B} \frac{\partial \theta_{c,j}^i}{\partial \theta_{c,j-1}^i} \right] \cdot \theta_{c,0}^i$$

$$= \frac{\partial \mathcal{L}_m(\mu_l = 0; \mathcal{D}_B)}{\partial \theta_{c,B}^i} \cdot \left[ \prod_{j=1}^{B} \frac{\partial \theta_{c,j-1}^i - \epsilon \frac{\partial \mathcal{L}_m(\mu_l=0; \mathcal{D}_{j-1})}{\partial \theta_{c,j-1}^i}}{\partial \theta_{c,j-1}^i} \right] \cdot \theta_{c,0}^i \tag{17}$$

$$= \frac{\partial \mathcal{L}_m(\mu_l = 0; \mathcal{D}_B)}{\partial \theta_{c,B}^i} \cdot \left[ \prod_{j=1}^{B} \mathbf{I} - \epsilon \frac{\partial^2 \mathcal{L}_m(\mu_l = 0; \mathcal{D}_{j-1})}{\partial (\theta_{c,j-1}^i)^2} \right] \cdot \theta_{c,0}^i,$$

where $\mathbf{I}$ represents the identity matrix, and $\partial^2$ represents the second-order derivative. By dropping the terms with the sufficiently small learning rate $\epsilon$ inspired by [9], the approximation is as follows:

$$\frac{\partial \mathcal{L}_m(\mu_l = 0; \mathcal{D}_B)}{\partial \theta_{c,0}^i} \cdot \theta_{c,0}^i = \frac{\partial \mathcal{L}_m(\mu_l = 0; \mathcal{D}_B)}{\partial \theta_{c,B}^i} \cdot \left[ \prod_{j=1}^{B} \mathbf{I} - \epsilon \frac{\partial^2 \mathcal{L}_m(\mu_l = 0; \mathcal{D}_{j-1})}{\partial (\theta_{c,j-1}^i)^2} \right] \cdot \theta_{c,0}^i$$

$$\approx \frac{\partial \mathcal{L}_m(\mu_l = 0, \mathcal{D}_B)}{\partial \theta_{c,B}^i} \cdot \left[ \prod_{j=1}^{B} \mathbf{I} \right] \cdot \theta_{c,0}^i \tag{18}$$

$$= \frac{\partial \mathcal{L}_m(\mu_l = 0, \mathcal{D}_B)}{\partial \theta_{c,B}^i} \cdot \theta_{c,0}^i.$$

And we finally prove that

$$\frac{\partial \mathcal{L}_m(\mu_l = 0; \mathcal{D}_B)}{\partial \theta_{c,0}^i} \cdot \theta_{c,0}^i \approx \frac{\partial \mathcal{L}_m(\mu_l = 0, \mathcal{D}_B)}{\partial \theta_{c,B}^i} \cdot \theta_{c,0}^i. \tag{19}$$

$$\square$$

# B  Detailed AlterEva for parameters in the LiDAR backbone

Due to the page limits, in Section 3.3, we only introduce the detailed formulation of the camera backbone. In this section, we will complete the details of the AlterEva of parameters from the LiDAR backbone, with two indicators DeCI and ReRI.

**Deactivated Contribution Indicator.** Similar to the parameters of camera backbones, the loss changes when parameters are deactivating via masking is observed, denoted as follows,

$$\hat{\Phi}_{\theta_l^i} = |\mathcal{L}_m(;\mathcal{D}) - \mathcal{L}_m(\mu_l^i = 0;\mathcal{D})|. \tag{20}$$

Due to the enormous number of parameters, we generalize this deactivating approach to encompass the entire LiDAR backbone. We then denote the new evaluation function as follows:

$$\hat{\Phi}_{\theta_l} = |\mathcal{L}_m(;\mathcal{D}) - \mathcal{L}_m(\mu_l = 0;\mathcal{D})|. \tag{21}$$

Then, to accommodate the need for differentiation among various parameters, we apply the Taylor first-order expansion to $\hat{\Phi}_{\theta_l}$ on each individual parameter $\theta_l^i$ in the camera backbone, considering $\theta_l = \{\theta_l^1, ..., \theta_c^{N_l}\}$.

$$\hat{\Phi}_{\theta_l^i} = \left| \mathcal{L}_m(;\mathcal{D}) + \mu_l^i \odot \theta_l^i \cdot \frac{\partial \mathcal{L}_m(;\mathcal{D})}{\partial \theta_l^i} - \mathcal{L}_m(\mu_l = 0;\mathcal{D}) - \mu_l^i \odot \theta_l^i \cdot \frac{\partial \mathcal{L}_m(\mu_l = 0;\mathcal{D})}{\partial \theta_l^i} \right|. \tag{22}$$

Since $\mu_l^i = 0$ when $\theta_l^i$ is deactivating, and constant loss values can be disregarded when considering importance scores on a global scale, the final indicator of each parameter's contribution, represented by our DeCI, can be expressed as follows:

$$\hat{\Phi}_{\theta_l^i} = \left| \theta_l^i \cdot \frac{\partial \mathcal{L}_m(;\mathcal{D})}{\partial \theta_l^i} \right|. \tag{23}$$

**Reactivated Redundancy Indicator.** Similarly, aiming to reactivate fusion-redundant parameters in the camera backbone, the camera backbone will first be masked with $\mu_c = 0$ and then masked models are trained with sampled batches. Throughout this process, the loss evolves from $\mathcal{L}_m(\mu_c = 0;\mathcal{D}_1)$ to $\mathcal{L}_m(\mu_c = 0;\mathcal{D}_B)$, and the parameters evolve from $\theta_{l,0}$ (i.e. $\theta_l$) to $\theta_{l,B}$. The loss changes and parameter differences are observed via our ReRI, denoted as follows:

$$\begin{aligned} \tilde{\Phi}_{\theta_l} &= |\mathcal{L}_m(\mu_c = 0;\mathcal{D}) - \mathcal{L}_m(\mu_c = 0;\mathcal{D}_1) + ... + \mathcal{L}_m(;\mathcal{D}_{B-1}) - \mathcal{L}_m(\mu_c = 0;\mathcal{D}_B)| \\ &= |\mathcal{L}_m(\mu_c = 0;\mathcal{D}) - \mathcal{L}_m(\mu_c = 0;\mathcal{D}_B)|. \end{aligned} \tag{24}$$

Then, we apply the first-order Taylor expansion to the initial $i$-th parameters $\theta_{l,0}^i$, denoted as:

$$\begin{aligned} \tilde{\Phi}_{\theta_{l,0}^i} = \Big| &\mathcal{L}_m(\mu_l = 0;\mathcal{D}) + \mu_l^i \odot \theta_{l,0}^i \cdot \frac{\partial \mathcal{L}_m(\mu_c = 0;\mathcal{D})}{\partial \theta_{l,0}^i} \\ &- \mathcal{L}_m(\mu_l = 0;\mathcal{D}_B) - \mu_l^i \odot \theta_{l,0}^i \cdot \frac{\partial \mathcal{L}_m(\mu_c = 0;\mathcal{D}_B)}{\partial \theta_{l,0}^i} \Big|. \end{aligned} \tag{25}$$

According to Proposition A, we eliminate the identical parts and apply the known value of $\mu_c = 1$, and the final formulation could be denoted as,

$$\tilde{\Phi}_{\theta_l^i} = \left| \theta_l^i \cdot \frac{\partial \mathcal{L}_m(\mu_c = 0;\mathcal{D})}{\partial \theta_l^i} - \theta_l^i \cdot \frac{\partial \mathcal{L}_m(\mu_c = 0;\mathcal{D}_B)}{\partial \theta_{l,B}^i} \right|. \tag{26}$$

Therefore, with the combination Eqn. 23 and Eqn. 26, the final importance scores evaluation function AlterMOMA of the LiDAR backbones could be presented with a normalization:

$$\mathcal{S}(\theta_l^i) = \alpha \cdot \frac{\hat{\Phi}_{\theta_l^i}}{\sum_{j=0}^{N_l} \hat{\Phi}_{\theta_l^j}} - \beta \cdot \frac{\tilde{\Phi}_{\theta_l^i}}{\sum_{j=0}^{N_l} \tilde{\Phi}_{\theta_l^j}}. \tag{27}$$

# C  Detailed AlterEva for parameters in the Fusion Modules and Following Task Heads

Similar with Appendix B, in this section, we will complete the details of the AlterEva of parameters from the fusion backbone, with two indicators DeCI and ReRI.

**Deactivated Contribution Indicator.** Similar to the parameters of the camera and LiDAR backbones, the loss changes when parameters are deactivating via masking is observed and then expand with a Taylor first-order expansion, denoted as follows,

$$\hat{\Phi}_{\theta_f^i} = \left| \mathcal{L}_m(;\mathcal{D}) + \mu_f^i \odot \theta_f^i \cdot \frac{\partial \mathcal{L}_m(;\mathcal{D})}{\partial \theta_f^i} - \mathcal{L}_m(\mu_f = 0;\mathcal{D}) - \mu_f^i \odot \theta_f^i \cdot \frac{\partial \mathcal{L}_m(\mu_f = 0;\mathcal{D})}{\partial \theta_f^i} \right|. \tag{28}$$

Then, after simplifying the constant loss and masked (deactivated) terms, formulation DeCI can be expressed as follows:

$$\hat{\Phi}_{\theta_f^i} = \left| \theta_f^i \cdot \frac{\partial \mathcal{L}_m(;\mathcal{D})}{\partial \theta_f^i} \right|. \tag{29}$$

**Reactivated Redundancy Indicator.** Different from the above camera backbone formulation in Section 3.3 and LiDAR backbone formulation in Appendix B, the fusion backbone experience the both alternative masking stages of AlterEva. Therefore, the ReRI of parameters in fusion modules and following task heads are calculated twice, denoted as,

$$\tilde{\Phi}_{\theta_f}(\mu_c = 0) = |\mathcal{L}_m(\mu_c = 0;\mathcal{D}) - \mathcal{L}_m(\mu_c = 0;\mathcal{D}_B)|, \tag{30}$$

$$\tilde{\Phi}_{\theta_f}(\mu_f = 0) = |\mathcal{L}_m(\mu_f = 0;\mathcal{D}) - \mathcal{L}_m(\mu_f = 0;\mathcal{D}_B)|, \tag{31}$$

Specifically, when loss evolves from $\mathcal{L}_m(\mu_c = 0;\mathcal{D}_1)$ to $\mathcal{L}_m(\mu_c = 0;\mathcal{D}_B)$ with masking camera backbone, the parameters evolve from $\theta_{f,0,\mu_c=0}$ (i.e. $\theta_f$)) to $\theta_{f,B,\mu_c=0}$. Meanwhile, when loss evolves from $\mathcal{L}_m(\mu_l = 0;\mathcal{D}_1)$ to $\mathcal{L}_m(\mu_l = 0;\mathcal{D}_B)$ with masking backbone, the parameters evolve from $\theta_{f,0,\mu_l=0}$ (i.e. $\theta_f$) to $\theta_{f,B,\mu_l=0}$. Then, we apply the first-order Taylor expansion to the initial $i$-th parameters $\theta_{l,0}^i$, denoted as:

$$\tilde{\Phi}_{\theta_f^i}(\mu_c = 0) = \left| \mathcal{L}_m(\mu_c = 0;\mathcal{D}) + \mu_f^i \odot \theta_{f,0,\mu_c=0}^i \cdot \frac{\partial \mathcal{L}_m(\mu_c = 0;\mathcal{D})}{\partial \theta_{f,0,\mu_c=0}^i} \right.$$
$$\left. - \mathcal{L}_m(\mu_c = 0;\mathcal{D}_B) - \mu_f^i \odot \theta_{f,0,\mu_c=0}^i \cdot \frac{\partial \mathcal{L}_m(\mu_c = 0;\mathcal{D}_B)}{\partial \theta_{f,0,\mu_c=0}^i} \right|, \tag{32}$$

$$\tilde{\Phi}_{\theta_f^i}(\mu_l = 0) = \left| \mathcal{L}_m(\mu_l = 0;\mathcal{D}) + \mu_f^i \odot \theta_{f,0,\mu_l=0}^i \cdot \frac{\partial \mathcal{L}_m(\mu_l = 0;\mathcal{D})}{\partial \theta_{f,0,\mu_l=0}^i} \right.$$
$$\left. - \mathcal{L}_m(\mu_l = 0;\mathcal{D}_B) - \mu_f^i \odot \theta_{f,0,\mu_l=0}^i \cdot \frac{\partial \mathcal{L}_m(\mu_l = 0;\mathcal{D}_B)}{\partial \theta_{f,0,\mu_l=0}^i} \right|. \tag{33}$$

According to Proposition A, we eliminate the identical parts and apply the known value of $\mu_c = 1$, and the final formulation could be denoted as,

$$\tilde{\Phi}_{\theta_f^i}(\mu_c = 0) = \left| \theta_f^i \cdot \frac{\partial \mathcal{L}_m(\mu_c = 0;\mathcal{D})}{\partial \theta_f^i} - \theta_f^i \cdot \frac{\partial \mathcal{L}_m(\mu_c = 0;\mathcal{D}_B)}{\partial \theta_{f,B,\mu_c=0}^i} \right|, \tag{34}$$

$$\tilde{\Phi}_{\theta_l^i}(\mu_l = 0) = \left| \theta_f^i \cdot \frac{\partial \mathcal{L}_m(\mu_l = 0;\mathcal{D})}{\partial \theta_f^i} - \theta_f^i \cdot \frac{\partial \mathcal{L}_m(\mu_l = 0;\mathcal{D}_B)}{\partial \theta_{f,B,\mu_l=0}^i} \right|. \tag{35}$$

Specifically, since the ReRI are calculated twice for parameters in fusion models, we using the hyperparameters $(\beta/2)$ to control the normalization scale of AlterMOMA of fusion modules. Formally, with the combination Eqn. 29 and Eqn. 34, the final importance scores evaluation function AlterMOMA of the LiDAR backbones could be presented with a normalization:

$$\mathcal{S}(\theta_f^i) = \alpha \cdot \frac{\hat{\Phi}_{\theta_f^i}}{\sum_{j=0}^{N_f} \hat{\Phi}_{\theta_f^j}} - \frac{\beta}{2} \cdot \frac{\tilde{\Phi}_{\theta_f^i}(\mu_l = 0)}{\sum_{j=0}^{N_f} \tilde{\Phi}_{\theta_f^j}(\mu_l = 0)} - \frac{\beta}{2} \cdot \frac{\tilde{\Phi}_{\theta_f^i}(\mu_c = 0)}{\sum_{j=0}^{N_f} \tilde{\Phi}_{\theta_f^j}(\mu_c = 0)}. \tag{36}$$

# D  Pseudo Code of AlterMOMA

The overview of our framework AlterMOMA and the importance scores evaluation function AlterEva are respectively introduced in Section 3.2 and Section 3.3. Specifically, AlterMOMA employs

**Algorithm 1** Alternative Modality Masking Pruning

**Input:** Pruning ratio $\rho$; iteration steps n; Networks $\{\mathbf{F}_l, \mathbf{F}_c, \mathbf{F}_f\}$; Related parameters $\{\theta_l, \theta_c, \theta_f\}$; Dataset $\mathcal{D}$

1: Initialise binary mask $\mu_c, \mu_l$; $\theta_c = \mu_c \odot \theta_c^{init}$, $\theta_l = \mu_l \odot \theta_l^{init}$.  ▷ Initialise Masks
2: **for** $m \in \{l, c\}$ **do**
3:     $\mu_m \leftarrow 0$  ▷ Alternative Modality Masking
4:     Train Masked Model with sampled batches from $\mathcal{D}_1$ to $\mathcal{D}_B$  ▷ Redundancy Reactivation
5:     Update importance scores with AlterEva by Eqn 11, 12 and 13  ▷ Importance Evaluation
6:     $\theta \leftarrow \theta^{init}$  ▷ Reinitialization
7: **end for**
8: Threshold $\tau \leftarrow (1 - \rho)$ percentile of $\mathcal{S}(\theta)$  ▷ Set pruning threshold
9: $\mu \leftarrow (\tau \leq \mathcal{S}(\theta))$  ▷ Set pruning mask
10: $\theta = \mu \odot \theta^{init}$  ▷ Apply mask on initial parameters
11: Finetune models with Dataset $\mathcal{D}$  ▷ Train Pruned model

Table 6: **3D object detection performance comparison with the state-of-the-art pruning methods** on the KITTI Validation dataset on Car class. We list the models pruned by different approaches within 80%, and 90% pruning ratios. The baseline model is trained with AVOD-FPN architecture.

| Sparsity | 80% | | | | | | 90% | | | | | |
|---|---|---|---|---|---|---|---|---|---|---|---|---|
| Tasks | AP-3D | | | AP-BEV | | | AP-3D | | | AP-BEV | | |
| Sparsity | Easy | Moderate | Hard | Easy | Moderate | Hard | Easy | Moderate | Hard | Easy | Moderate | Hard |
| Car [No Pruning] | 82.4 | 72.2 | 66.5 | 89.4 | 83.9 | 78.7 | - | - | - | - | - | - |
| IMP | 65.8 | 57.7 | 51.3 | 69.2 | 64.6 | 59.7 | 52.1 | 45.7 | 43.2 | 59.6 | 54.2 | 51.7 |
| SynFlow | 74.2 | 65.7 | 60.2 | 79.5 | 75.3 | 70.3 | 64.5 | 54.7 | 48.1 | 73.5 | 67.6 | 64.4 |
| SNIP | 73.5 | 64.9 | 59.8 | 79.1 | 75.8 | 69.6 | 62.7 | 52.3 | 45.8 | 72.4 | 66.9 | 63.7 |
| ProPr | 78.9 | 69.6 | 62.1 | 85.2 | 79.1 | 75.7 | 74.2 | 63.4 | 59.1 | 81.2 | 75.1 | 71.9 |
| AlterMOMA (Ours) | 80.5 | 70.2 | 63.2 | 87.2 | 81.5 | 77.9 | 77.4 | 68.2 | 62.3 | 85.3 | 79.9 | 75.8 |
| Pedestrian [No Pruning] | 50.5 | 43.2 | 40.1 | 58.1 | 50.7 | 47.2 | - | - | - | - | - | - |
| IMP | 34.3 | 27.8 | 23.5 | 40.9 | 32.5 | 29.9 | 28.7 | 20.2 | 16.4 | 34.1 | 29.2 | 25.8 |
| SynFlow | 43.7 | 36.3 | 32.3 | 50.2 | 45.1 | 41.5 | 33.9 | 25.4 | 20.6 | 43.5 | 38.9 | 35.6 |
| SNIP | 43.4 | 35.5 | 31.8 | 49.5 | 44.9 | 38.6 | 32.7 | 23.7 | 19.8 | 41.9 | 37.4 | 34.1 |
| ProPr | 46.9 | 38.1 | 34.7 | 54.9 | 49.4 | 44.2 | 43.8 | 36.9 | 33.5 | 51.2 | 44.7 | 39.9 |
| AlterMOMA (Ours) | 49.4 | 41.2 | 37.3 | 57.0 | 49.7 | 46.5 | 47.8 | 39.6 | 36.1 | 55.9 | 46.2 | 43.2 |
| Cyclist [No Pruning] | 63.8 | 51.7 | 45.2 | 67.6 | 57.2 | 50.4 | - | - | - | - | - | - |
| IMP | 47.2 | 36.8 | 30.7 | 51.2 | 41.4 | 35.7 | 39.2 | 31.5 | 27.3 | 45.5 | 36.9 | 31.2 |
| SynFlow | 56.2 | 44.5 | 36.5 | 59.1 | 48.6 | 42.2 | 47.5 | 36.2 | 31.1 | 54.8 | 44.3 | 38.2 |
| SNIP | 55.5 | 43.1 | 36.2 | 58.4 | 47.5 | 41.9 | 47.2 | 35.4 | 29.2 | 53.3 | 42.3 | 37.5 |
| ProPr | 60.0 | 47.5 | 41.6 | 62.7 | 52.2 | 46.1 | 57.4 | 45.2 | 39.6 | 60.3 | 49.9 | 43.7 |
| AlterMOMA (Ours) | 62.1 | 49.8 | 44.9 | 65.2 | 54.8 | 48.3 | 59.7 | 47.9 | 43.1 | 62.5 | 52.3 | 46.0 |

alternative masking on each modality, followed by the observation of loss changes when certain modality parameters are activated and deactivated. These observations serve as important indications to identify fusion-redundant parameters, which are integral to our importance scores evaluation function, AlterEva. AlterMOMA begins with *Modality Masking*, where one of the backbones is initially masked. This step is followed by *Redundancy Reactivation* and *Importance Evaluation*, where the parameter importance scores are initially calculated with AlterEva including the computation of DeCI and ReRI. Afterward, the models undergo *Reinitialization* and *Alternative Masking* of the other backbone, leading to another round of *Redundancy Reactivation* and *Importance Evaluation*. When scores of all parameters in backbones are calculated fully with AlterEva, models are pruned to remove parameters with low importance scores and then finetuned.

We have also extended AlterMOMA to accommodate structured pruning, where instead of pruning individual weights, entire channels (or columns in linear layers) are removed. Although this approach imposes more restrictions, it significantly enhances memory efficiency and reduces the computational costs associated with training and inference. Adapting AlterMOMA to structured pruning involves simply modifying the shape of the pruning mask $\mu$ and the parameter $\theta$ in formulation to represent each channel (or column of the weight matrix). For experimental results, please refer to Section 4.4.

# E   Complete Experimental Results of KITTI dataset

As detailed in Section 4.1, we conducted experiments using AVOD-FPN [3] on the KITTI dataset [17] to demonstrate the efficacy of our AlterMOMA on two-stage fusion architectures. Due to space constraints, we initially presented only a subset of our results on the KITTI dataset in Section 4.3, specifically excluding the results for the Pedestrian and Cyclist classes. In this section, we aim to

Table 7: **3D object detection performance comparison with camera-radar fusion models** on the nuScenes validation dataset. The baseline model is trained with ResNet and PointPillars backbone.

| Sparsity | 80% | | 90% | |
|---|---|---|---|---|
| | mAP | NDS | mAP | NDS |
| **BEVfusion-R [No Pruning]** | 40.3 | 50.1 | - | - |
| ProPr | 35.8 | 43.6 | 32.4 | 40.1 |
| **AlterMOMA (Ours)** | **38.5** | **48.3** | **35.2** | **44.3** |

Table 8: **3D multi-object tracking (MOT) task performance comparison** by performing tracking-by-detection on the nuScenes validation dataset. We list the AMOTA of models pruned within 80% and 90% pruning ratios. The baseline model is trained with SwinT and VoxelNet backbone.

| Sparsity | 80% | 90% |
|---|---|---|
| | AMOTA | AMOTA |
| **BEVfusion-mit [No Pruning]** | 68.2 | - |
| ProPr | 65.2 | 61.4 |
| **AlterMOMA (Ours)** | **67.1** | **64.5** |

provide a comprehensive view of our findings on KITTI. The complete experimental results are displayed in Table 6. Specifically, Table 6 discloses results for all classes (Car, Pedestrian, and Cyclist) on the KITTI dataset, detailing AP-3D and AP-BEV accuracy across varying difficulty levels, including *easy*, *moderate*, and *hard*. As reported in Table 6, single-modal pruning methods, including IMP, SynFlow, SNIP, and ProPr, experience significant declines in accuracy performance in all classes. Conversely, the incorporation of our AlterMOMA yielded promising results.

# F    Extended Experimental Results on Camera-Radar Models

While our primary focus has been on camera and LiDAR modalities, we recognize that testing on additional modalities would make our proposed method more convincing. Therefore, we conducted further evaluations on the camera-radar modality, as presented in Table 7. This includes a comparison of pruning results on the 3D object detection task using BEVFusion-R, a camera-radar fusion model with ResNet and PointPillars as backbones. We list the mAP and NDS of models at 80% and 90% pruning ratios. Compared to the baseline pruning method ProsPr, AlterMOMA boosts the mAP of BEVFusion-R by 2.7% and 2.8% for the two different pruning ratios. The results demonstrate our method's superior performance and generality across multiple modalities, including camera-radar.

# G    Extended Experimental Results on Tracking Tasks

To further validate the generalizability of our method across different tasks, we conducted additional evaluations on multi-object tracking (MOT), a critical task in autonomous driving, as shown in Table 8. We performed tracking-by-detection evaluation on the nuScenes validation dataset, and list the AMOTA of models pruned at 80% and 90% pruning ratios. The baseline model was trained with SwinT and VoxelNet backbones. Compared to the baseline pruning method ProsPr, AlterMOMA boosts the AMOTA of BEVFusion-mit by 1.9% and 3.1% for the two pruning ratios. These results further demonstrate our method's superior performance and generality across multiple tasks, including tracking.

# H    Inference Speed of AlterMOMA

In this section, we report the running time in milliseconds in Table 9. We tested the inference time of models after pruning using the structure pruning settings of the Table 5 of the main paper. Here are the performance results for BEVFusion-mit models with SECOND and ResNet101 backbones on an single RTX 3090: the inference times are 124.04 ms for unpruned models, 106.39 ms for AlterMOMA-30%, and 87.46 ms for AlterMOMA-50%. These results highlight the improvements in inference speed achieved through our pruning techniques.

Table 9: **3D object detection performance and inference speed comparison with the structure pruning methods** on the nuScenes validation dataset. Note that baseline model is BEVfusion-mit with ResNet101 and SECOND as backbone and inference is tested on the RTX3090.

| Sparsity | ResNet101 + SECOND | | | |
|---|---|---|---|---|
| | mAP | NDS | GFLOPs($\downarrow$ %) | Inference time(ms) |
| **BEVfusion-mit [No Pruning]** | 64.6 | 69.4 | 610.66 | 124.04 |
| **AlterMOMA-30% (Ours)** | **65.3** | **69.9** | **420.13 (31.2)** | 106.39 |
| **AlterMOMA-50% (Ours)** | **64.5** | **69.5** | **264.42 (56.7)** | 87.46 |

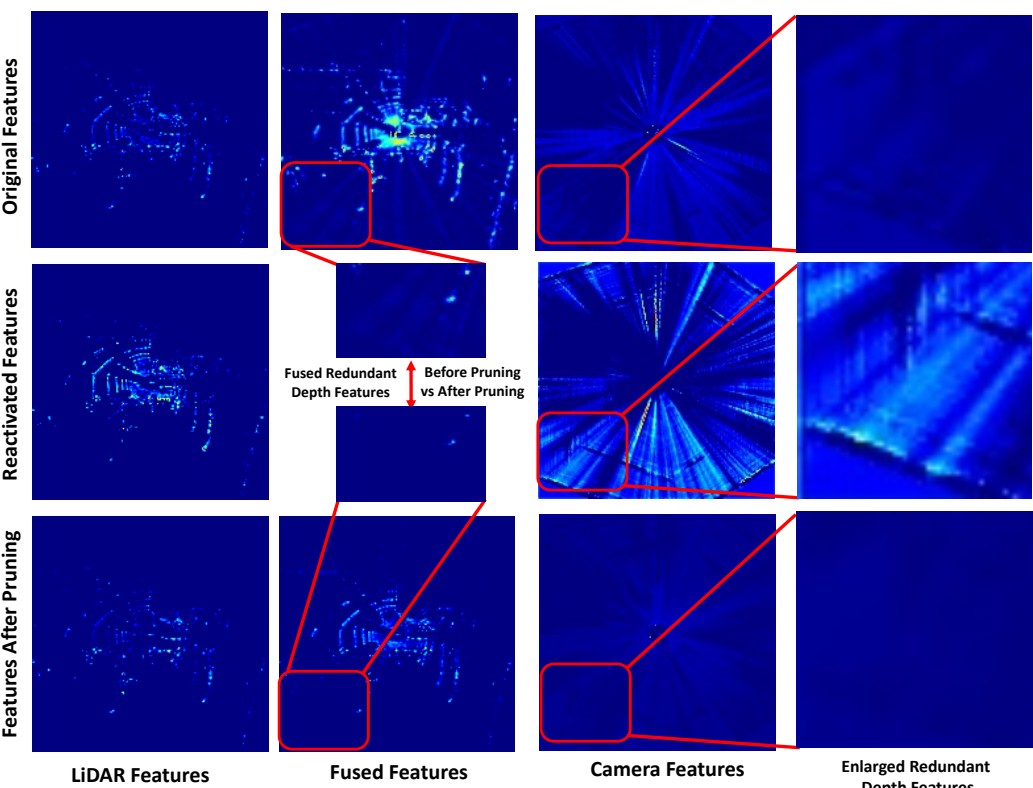

Figure 3: **Visualization of the reactivated redundant features**: The figure illustrates the features of different modalities at each stage of the entire pruning process of AlterMOMA, including LiDAR features, camera features and fused features in the states of original (before masking), reactivated (after reactivation), and pruned (after pruning with AlterMOMA).

# I Analysis and Visualization of Features

The Figure 3 illustrates the features of different modalities at each stage of the entire pruning process of AlterMOMA, including LiDAR features, camera features and fused features in the states of original (before masking), reactivated (after reactivation), and pruned (after pruning with AlterMOMA). The fourth column provides enlarged views of crucial redundant parts of the camera features. Notably, due to masking one side of backbones during reactivation, there are no fused features within reactivated states. Specifically, despite the absence of distant objects, camera features still provide some redundant depth features, as shown in the fourth column. These redundant parts of the original camera features are retained in the subsequent original fused features after fusion. To address these redundant depth features, AlterMOMA reactivates these redundant parameters during the reactivation phase, as shown in the middle row images of the fourth column. These redundant depth features are then pruned from both fused features and camera features, as observed by comparing the enlarged fused features in the middle row images of the second column and the pruned camera backbone features in the third and fourth columns of the third row.

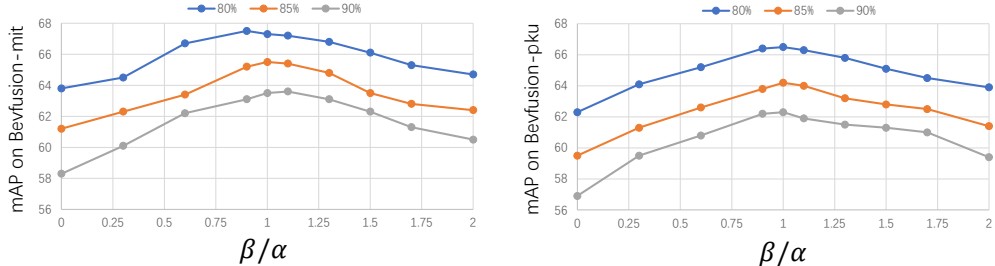

Figure 4: **Ablation study** of hyperparameters $\alpha$ and $\beta$ on the nuScenes validation dataset. We list the relationship between mAP and $\beta/\alpha$ with our approaches within 80%, 85%, and 90% pruning ratios. The two baseline models, BEVfusion-mit and BEVfusion-pku are trained with SwinT and VoxelNet backbone.

## J    Ablation Study on hyper parameters $\alpha$ and $\beta$

As described in Section 3.3, the hyperparameters $\alpha$ and $\beta$ determine the proportion of DeCI and ReRI within the importance score evaluation function AlterEva. To assess the impact of these indicators on overall pruning performance, we examine the relationship between mAP and the ratio $\beta/\alpha$. Baseline models depicted in the left subfigures of Figure 4 are from BEVfusion-mit trained with SwinT and VoxelNet backbones, while the right subfigures represent BEVfusion-pku models trained with the same backbones. These experiments utilize the nuScenes dataset to evaluate the efficiency of 3D object detection tasks at pruning ratios of 80%, 85%, and 90%. The experimental results are presented in Figure 4. Specifically, when $\beta/\alpha = 0$ in the figure, indicating only relying on DeCI, there is a significant drop in mAP compared to our best-performing setup. This result underscore the critical role of addressing fusion-redundant parameters. As the ratio increases, indicating greater influence from ReRI, mAP increases, reflecting the beneficial impact of effectively identifying and pruning fusion-redundant parameters. However, as $\beta$ surpasses a certain threshold, resulting in ReRI outweighing DeCI, mAP begins to decline again. It may be due to that our redundancy reactivation also reactivates some contributed parameters, which may be accidentally pruned with the excessive usage of ReRI. These findings highlight the selection of hyper parameters and the ablation study for DeCI and ReRI indicators in AlterMOMA. The results are presented in Figure 4, which visually depicts these dynamics across different experimental settings.

## K    Further Discussion of AlterMOMA on General Multi-modal Fusion models

Although AlterMOMA explores similar feature extraction due to the fusion mechanism, it remains in the perception fields, especially camera-LiDAR fusion models. However, extending it to a wider range of multi-modal models, such as vision-language models, requires further refinement. As we hypothesize, fusion modules across various modalities and tasks exhibit different functionalities. In perception-only tasks involving multiple sensors (camera, LiDAR and radar), the fusion mechanism primarily focuses on supplementing and spatial aligning across modalities by fully leveraging the physical properties of different sensors since all inputs consist of vision-based data. Formally, low-level machine features are fused in the multi-sensor fusion mechanism. However, in the fusion mechanism devised for different types of data, such as vision and language, things differ. When input data is highly disparate, such as vision and language, fusion modules tend to focus more on matching, which means synchronizing the high-level semantic context between different input formats. Therefore, for the AlterMOMA framework, we concentrate on the redundancy stemming from supplementary functionality, which may only primarily exist in multi-sensor fusion architectures.

