# OpenReview forum: "AlterMOMA: Fusion Redundancy Pruning for Camera-LiDAR Fusion Models with Alternative Modality Masking"
_NeurIPS.cc/2024/Conference — NeurIPS 2024 poster_

### Official Review · Reviewer_62BJ · 2024-06-24

**Soundness:** 3
**Presentation:** 4
**Contribution:** 4
**Rating:** 7
**Confidence:** 3

**Summary:**

This paper introduces Alternative Modality Masking Pruning (AlterMOMA), a pruning framework utilizing alternative masking on each modality to pinpoint redundant parameters. It detects redundant features and relevant parameters through the reactivation process. These redundant parameters are then pruned using a proposed importance score evaluation function. Meanwhile, the observation of loss changes indicates which modality parameters are activated or deactivated.

**Strengths:**

1. This paper is well-written, and the concept of alternately applying masking to the camera and Lidar backbones to identify and eliminate fusion-redundant parameters is innovative.
2. When designing the Deactivated Contribution Indicator (DeCI) and Reactivated Redundancy Indicator (ReRI) modules, thorough analysis and formula derivation were conducted for loss change processes using the masking method.
3. Extensive experiments on the nuScenes and KITTI datasets have demonstrated the effectiveness of AlterMOMA.

**Weaknesses:**

The paper lacks an illustration of the costs associated with the proposed pruning method itself.

**Questions:**

How many rounds does "Modal masking-Redundant activation-importance score evaluation" take?

**Limitations:**

Experiments indicate that the proposed Pruning method is effective. However, the effectiveness of other multi-modal fusion methods, such as VLM, still needs to be verified.

---

> ### Author Rebuttal · Authors · 2024-08-07
>
> Thank you for your positive feedback and insightful comments. We will answer your questions in the following:
>
> ***Weakness 1***:  *Lack of the computataional overhead of pruning process.*
>
>  - Thank you for reminding us of the importance of detailing the cost associated with the proposed pruning method, including the execution process of algorithms and the rounds count of the entire pruning phases. We agree that providing this information will significantly enhance the quality of this paper and we will add them in the paper. Therefore, we present the detailed computation cost as follows:
> The pruning process involves **one round** of *"Masking (LiDAR) - Reactivation (Camera) - AlterMasking (Camera) - Reactivation (LiDAR) - Importance evaluation"* before the beginning of each fine-tuning epochs, and the total **fine-tuning epoch number is 3**. In each 'Reactivation' stage, we only sample **100 out of 62k batches** (the entire nuScenes training set comprises 62k batches with a batch size of 2). We tested the time for **the reactivation stage, which is around 170 seconds**. In the Importance evaluation stage, the important scores for parameters were evaluated by using the gradients via backpropagation before and after reactivation for 5 batches, as mentioned in Eqn.11-13 of the main paper. We tested the time for **the importance evaluation stage, which is around 10 seconds**. We tested the time for **the entire round, which is an average of around 190 seconds** in total on a single RTX3090 device. Subsequently, a 3 epochs fine-tuning is performed, which incurs the same computational cost as normal training.
> Please note that unpruned baseline models, such as BEVFusion-mit, require a total of 30 epochs to train from scratch, including 20 epochs for LiDAR backbone pre-training and 10 epochs for the entire fusion architecture training. Since pruning is a one-time expense for generating a compact model, its overhead can be considered trivial, especially when considering the time-consuming nature of normal training. Due to the page limit, the aforementioned discussion will be included in the supplementary material for the camera-ready version.
>
> ***Question 1***:  *How many rounds does "Modal masking-Redundant activation-Importance score evaluation" take?*
>
> - We appreciate the professional question. Regarding the cost of the pruning methods in weakness 1, the process involves **one round** of *"Masking (LiDAR) - Reactivation (Camera) - AlterMasking (Camera) - Reactivation (LiDAR) - Importance evaluation"* before the beginning of each fine-tuning epochs, and the total fine-tuning epoch number is 3. Please let us know if any further adjustments or additional details are required.
>
> ***Limitation:*** *However, the effectiveness of other multi-modal fusion methods, such as VLM, still needs to be verified.*
>
> - We appreciate the insightful and inspiring comment. We currently focus on camera and LiDAR modalities, because these modalities are the most important factors for the perception of autonomous driving. We agree that testing on more modalities would make the proposed method more convincing. Therefore, we have performed additional evaluations on the **Camera-Radar modality in Table 1 of the uploaded global rebuttal PDF file**. Regarding the vision-language fusion methods such as VLM, we are working on pruning the visual grounding task using natural language and camera modalities. We have added a related discussion in the supplementary material of the main paper and will share our findings as soon as we obtain promising results.

---

> > ### Comment · Reviewer_62BJ · 2024-08-13
> >
> > Thanks for the authors' rebuttals, the illustration is clear. I will keep my initial score of 7 (Accept).

---

> > > ### Author Response · Authors · 2024-08-13
> > >
> > > Thank you for your insightful reviews and for maintaining your initial score. We will address your feedback and include the analysis of the computational overhead of the pruning process in the further camera-ready version. We greatly appreciate your valuable insights and the time you’ve dedicated to evaluating our work.

---

### Official Review · Reviewer_fjuq · 2024-07-08

**Soundness:** 3
**Presentation:** 3
**Contribution:** 3
**Rating:** 6
**Confidence:** 3

**Summary:**

This paper addresses the problem of feature redundancy in camera-LiDAR fusion models. The authors propose a novel pruning framework, AlterMOMA, which employs alternative masking to identify and prune redundant parameters in these models. The paper demonstrates the effectiveness of AlterMOMA through extensive experiments on the nuScenes and KITTI datasets, showing superior performance compared to existing pruning methods.

**Strengths:**

1. The paper introduces a novel pruning framework specifically designed for multi-modal fusion models, addressing the unique challenge of redundant features across different modalities
2. Extensive results on nuScenes and KITTI showcasing the benefits and performance improvements of the proposed approach
3. The paper is very well written and easy to follow

**Weaknesses:**

1. While the method is tested on two camera-LiDAR datasets, the generalizability of the approach to other fusion tasks and modalities beyond camera-LiDAR is not verified, such as video, optical flow and audio in action recognition [A].
2. The running speed after pruning is not reported
3. In Table 5, the performance after pruning is even better than the original version. Are there any reasons and analysis?

[A] Munro et al., Multi-modal domain adaptation for fine-grained action recognition. In CVPR, 2020.

**Questions:**

Can the proposed method be effective on other tasks like action recognition?

**Limitations:**

The paper mentions one limitation regarding its limited use in the perception field.

---

> ### Author Rebuttal · Authors · 2024-08-07
>
> Thanks for your insightful comments and suggestions. We will address your concerns in the following answers:
>
> ***Weakness 1:*** *Generalizability of AlterMOMA to additional modalities and tasks.*
> - We appreciate the inspiring comments. We currently focus on camera and LiDAR modalities, as well as 3D object detection and BEV map segmentation tasks, because these modalities and tasks are the most important factors for the perception of autonomous driving.
> We agree that testing on more modalities and tasks would make the proposed method more convincing. However, preparing new datasets, adapting our pruning frameworks to new fusion architectures, and training from scratch is time-consuming. We attempted to implement experiments on action recognition with [1], but it is challenging to accomplish this in the limited rebuttal period. Although we cannot test on action recognition tasks, we have expanded our models to more tasks and modalities to demonstrate our generalizability in autonomous driving. We have performed additional evaluations on the **Camera-Radar modality** and **the multi-object tracking (MOT) task** seperately in **Table 1 and Table 2 of uploaded global rebuttal PDF file**.
> **For the Camera-Radar modality, as presented in Table 1 of supplementary PDF file,** it includes a comparison of pruning results on the 3D object detection task using BEVFusion-R, a Camera-Radar fusion model with ResNet and PointPillars as backbones. We list the mAP and NDS of models at 80% and 90% pruning ratios. Compared to the baseline pruning method ProsPr, AlterMOMA boosts the mAP of BEVFusion-R by 2.7% and 2.8% for the two different pruning ratios. The results demonstrate our method’s superior performance and generality across multiple modalities, including Camera-Radar.
> **For the MOT task, as shown in Table 2 of supplementary PDF file,** we performed evaluation on the nuScenes validation dataset, and list the AMOTA of models pruned at 80% and 90% pruning ratios. Compared to the baseline pruning method ProsPr, AlterMOMA boosts the AMOTA of BEVFusion-mit by 1.9% and 3.1% for the two pruning ratios. These results further demonstrate our method's superior performance and generality across multiple tasks, including tracking.
> In future research, we plan to explore the applicability of AlterMOMA incorporating more tasks like action recognition. We appreciate your understanding of our current focus and look forward to exploring these avenues in future studies.
>
> ***Weakness 2:*** *The running speed after pruning.*
> - We follow SOTA papers and report the computational efficiency in terms of GFLOPs. We agree that additionally reporting running speed would make the paper more practical and convincing. Therefore, we additionally report the running time in milliseconds.  We have tested the inference time of BEVFusion-mit models after pruning with structure pruning setting of Table 5 of the main paper on a single RTX3090 device. The inference times are **124.04 ms for unpruned models, 106.39 ms for AlterMOMA-30%, and 87.46 ms for AlterMOMA-50%**. These results highlight the improvements in inference speed achieved through our pruning techniques. The result table are also presented **in Table 3 of  uploaded gobal rebuttal PDF file** and will be included in the camera-ready version.
>
> ***Weakness 3:*** *Analysis of outperformed results in Table 5.*
> - Thank you for reminding us of the need for further analysis on this positive result, which supports our observation on fusion redundancy and would make the paper in better form. Therefore, following analysis is performed and will be included in the camera-ready version:
> As discussed in Section 1, the use of pre-trained backbones results in similar feature extractions. With this similarity, low-quality similar features (e.g., depth features from the camera backbone) are also extracted and may be fused in the following fusion modules, thereby disturbing their high-quality counterparts during fusion (e.g., depth features from the LiDAR backbone). Therefore, when these low-quality redundant features and their related parameters are pruned using our proposed AlterMOMA (especially at low pruning ratios, which ensure the pruning of low-quality features is sufficient), overall performance improves during subsequent fine-tuning due to the elimination of redundant disturbances. We also **visualize the fusion redundancy in the uploaded global rebuttal PDF file**, which further explains why the elimination of low-quality features enhances performance.
> Notablely, better performance after pruning compared to the original version has also been observed in several state-of-the-art studies, such as Table 4 and 5 in [2], Table 3 in [3], and Table 1 and 2 in [4].
>
> ***Question:*** *The implement of AlterMOMA on action recognition.*
> - We sincerely agree that evaluating the proposed method on action recognition would greatly improve the generalizability of AlterMOMA. Currently this work focuses on perception tasks of autonomous driving, and applying the proposed method on the tasks other than perception tasks of autonomous driving (i.e. action recognition) needs further research. Since the limited rebuttal period, we could not adapt the proposed method and evaluate on action recognition. In order to response the question and enhance the generalizability of this paper on more tasks, we performed the proposed method on multi-object tracking tasks, which is the downstream task of 3D object detection and BEV map segmentation, which is detailed in **the answer of weakness 1 and also shown in the Table 1 of uploaded global rebuttal PDF file**.
>
> [1] Munro et al., Multi-modal domain adaptation for fine-grained action recognition. CVPR, 2020.
> [2] Huang Y et al. CP3: Channel pruning plug-in for point-based networks. CVPR 2023.
> [3] Lin M et al. Hrank: Filter pruning using high-rank feature map. CVPR, 2020.
> [4] Fang G et al. Depgraph: Towards any structural pruning. CVPR, 2023.

---

> > ### Comment · Reviewer_fjuq · 2024-08-11
> >
> > Thank the authors for their rebuttal. Most of my concerns have been well-addressed and I thus increased my score to 6.

---

> > > ### Author Response · Authors · 2024-08-12
> > >
> > > Thank you for considering our rebuttal and increasing our score. We sincerely appreciate your feedback and are grateful for your time in evaluating our work.

---

### Official Review · Reviewer_hcok · 2024-07-12

**Soundness:** 3
**Presentation:** 4
**Contribution:** 3
**Rating:** 7
**Confidence:** 3

**Summary:**

This paper introduces a novel and effective pruning framework (AlterMOMA) and outperforms existing pruning methods on multimodal 3D detection/segmentation tasks, based on a novel insight : "The absence of fusion-contributed features will compel fusion modules to ’reactivate’ their fusion-redundant counterparts as supplementary to maintain functionality".

**Strengths:**

- the paper presents a pioneering approach to pruning in camera-LiDAR fusion models, tackling redundancy through innovative masking strategies. The procedure is clearly explained and technically sound.
- the proposed method demonstrates good performance on multiple tasks on two baseline model.
- this paper proposes an interesting insight and tests it with experiments

**Weaknesses:**

- while AlterMOMA aims to reduce model complexity, the paper could clarify the computational overhead introduced by the pruning process itself, especially regarding the training time and resources required for reactivation and evaluation.
- it is interesting to use some more direct methods (e.g., feature visualization) to prove that the "reactivated" parameters are "redundant".
- some minor typing errors (e.g., line132, an incorrect quotation mark), which do not affect my rating

**Questions:**

the paper presents a pruning strategy for multimodal strategy, which seems to work on any multimodal task. However, the authors only experimented on the Camera-LiDAR task. Did authors try more modal combinations (Radar, or even Language, etc.)? Is the approach applicable to other broader tasks?

**Limitations:**

the paper discussed the limitations in the Appendix.

---

> ### Author Rebuttal · Authors · 2024-08-07
>
> We appreciate the professional and insightful comments. We address each comment as follows:
>
> ***Weakness 1:*** *The computataional overhead of pruning process.*
> - We agree that providing the computational overhead for the proposed pruning framework would significantly enhance the quality of this paper. Therefore, we present the computational expenses as follows:
> The pruning process involves **one round** of *"Masking (LiDAR) - Reactivation (Camera) - AlterMasking (Camera) - Reactivation (LiDAR) - Importance evaluation"* before the beginning of each fine-tuning epochs, and the **total fine-tuning epoch number is 3**. In each 'Reactivation' stage, we only **sample 100 out of 62k batches** (the entire nuScenes training set comprises 62k batches with a batch size of 2). We tested the time for **the reactivation stage, which is around 170 seconds**. In the Importance evaluation stage, the important scores for parameters were evaluated by using the gradients via backpropagation before and after reactivation for 5 batches, as mentioned in Eqn.11-13 of the main paper. We tested the time for **the importance evaluation stage, which is around 10 seconds**. We tested the time for **the entire round, which is an average of around 190 seconds** in total on a single RTX3090 device. Subsequently, a 3 epochs fine-tuning is performed, which incurs the same computational cost as normal training.
> Please note that unpruned baseline models, such as BEVFusion-mit, require a total of 30 epochs to train from scratch, including 20 epochs for LiDAR backbone pre-training and 10 epochs for the entire fusion architecture training. Since pruning is a one-time expense for generating a compact model, its overhead can be considered trivial, especially when considering the time-consuming nature of normal training.Due to the page limit, the aforementioned discussion will be included in the supplementary material for the camera-ready version.
>
> ***Weakness 2:*** *Feature visualization of reactivaed redundant parameters.*
> - We appreciate the insightful comment and acknowledge that adding feature visualizations would make our paper more convincing. Therefore, we have included feature visualization figures **in Figure 1 of the uploaded global rebuttal PDF file**. These figures will also be added to the camera-ready paper.
> The figure illustrates the features of different modalities at each stage of the entire pruning process of AlterMOMA, including LiDAR features, camera features and fused features in the states of original (before masking), reactivated (after reactivation), and pruned (after pruning with AlterMOMA). For better clarity and understanding, we enlarge views of crucial redundant parts of camera features in the fourth column of Figure 1. Notably, due to masking one side of backbones during reactivation, there are no fused features within reactivated states.
> Specifically, despite the absence of distant objects, camera features still provide some redundant depth features, as shown in the fourth column. These redundant parts of the original camera features are retained in the subsequent original fused features after fusion. To identify these redundant depth features, AlterMOMA reactivates these redundant parameters during the reactivation phase, as shown in the middle row images of the fourth column. These redundant depth features are then pruned from both fused features and camera features, as observed by comparing the enlarged fused features in the middle row images of the second column and the pruned camera backbone features in the third and fourth columns of the third row.
>
> ***Weakness 3:*** *Typing errors.*
> -  We appreciate the reminder about the typing errors. We have carefully proofread the entire paper and corrected the typographical errors, including the one on line 132.
>
> ***Question:*** *Generalizability of AlterMOMA to additional modalities and tasks.*
> - We appreciate the insightful and inspiring comment. We currently focus on camera and LiDAR modalities, as well as 3D object detection and BEV map segmentation tasks, because these modalities and tasks are the most important factors for the perception of autonomous driving. We agree that testing on more modalities and tasks would make the proposed method more convincing. Therefore, we have performed additional evaluations on the **Camera-Radar modality** and and the **multi-object tracking (MOT) task**  seperately **in Table 1 and Table 2 of uploaded global rebuttal PDF file**.
> **For the Camera-Radar modality, as presented in Table 1 of supplementary PDF file**, it includes a comparison of pruning results on the 3D object detection task using BEVFusion-R, a Camera-Radar fusion model with ResNet and PointPillars as backbones. We list the mAP and NDS of models at 80% and 90% pruning ratios. Compared to the baseline pruning method ProsPr, AlterMOMA boosts the mAP of BEVFusion-R by 2.7% and 2.8% for the two different pruning ratios. The results demonstrate our method’s superior performance and generality across multiple modalities, including Camera-Radar.
> **For the MOT task, as shown in Table 2 of supplementary PDF file**, we performed tracking-by-detection evaluation on the nuScenes validation dataset, and list the AMOTA of models pruned at 80% and 90% pruning ratios. The baseline model was trained with SwinT and VoxelNet backbones. Compared to the baseline pruning method ProsPr, AlterMOMA boosts the AMOTA of BEVFusion-mit by 1.9% and 3.1% for the two pruning ratios. These results further demonstrate our method's superior performance and generality across multiple tasks, including tracking.
> Please refer to the global rebuttal PDF file for more details. Regarding the language modality, we are working on pruning the visual grounding task using natural language and camera modalities. We have added a related discussion in the supplementary material of the main papaer and will share our findings as soon as we obtain promising results.

---

> > ### Comment · Reviewer_hcok · 2024-08-12
> >
> > The authors addressed most of my questions. It is surprising that the authors could add so many experiment results in the short rebuttal period. Based on this, I will maintain my initial score (7/Accept).
> >
> > Meanwhile, I would like to suggest the authors (1) make sure these additional experiment results can be added to the supplementary material, and (2) make sure the tables in the manuscript have the same/similar font size for better view (try not use \resizebox).

---

> > > ### Author Response · Authors · 2024-08-12
> > >
> > > Thank you for your thoughtful comments and for maintaining your initial score. We appreciate your recognition of the additional experiment results we included during the rebuttal period. We will ensure that these additional results are included in the supplementary material, and we will revise the tables in the manuscript to maintain a consistent font size for better readability, avoiding the use of \resizebox. We sincerely appreciate your feedback and are grateful for your time in evaluating our work.

---

### Author Rebuttal · Authors · 2024-08-07

Thank you to all the reviewers for your efforts in reviewing our work. Your insightful suggestions and comments are greatly appreciated and will significantly enhance the quality of our paper. We are grateful for the positive feedback on various aspects and are encouraged that the reviewers pointed out our work *"is clearly explained and technically sound"* and *"proposes an interesting insight"* (Reviewer hcok), *"introduces a novel pruning framework"* and *"very well written and easy to follow"* (Reviewer fjuq),  *"the concept of alternately applying masking is innovative"* and  *"extensive experiments"* (Reviewer 62BJ). We address the reviews below and will incorporate all changes in the revision.

Although we were unable to conduct exhaustive extended experiments on AlterMOMA due to the time-consuming nature of the process (e.g., preparing code environments, adapting our pruning frameworks to new model architectures, and training models from scratch on large-scale autonomous driving datasets), we have included **several additional experiments and visualizations in the supplementary PDF of global rebuttals.** These additions address common concerns and demonstrate the efficiency and generality of our work, including:

***1. The generalizability of AlterMOMA on additional modalities.***
- While our primary focus has been on camera and LiDAR modalities, we acknowledge that testing on additional modalities would make our proposed method more convincing. Therefore, we conducted further evaluations on the **Camera-Radar modality, as presented in Table 1 of supplementary PDF file.** This includes a comparison of pruning results on the 3D object detection task using BEVFusion-R, a Camera-Radar fusion model with ResNet and PointPillars as backbones. We list the mAP and NDS of models at 80% and 90% pruning ratios. Compared to the baseline pruning method ProsPr, AlterMOMA boosts the mAP of BEVFusion-R by 2.7% and 2.8% for the two different pruning ratios. The results demonstrate our method’s superior performance and generality across multiple modalities, including Camera-Radar.
 Regarding the language modality, we are working on pruning the visual grounding task using natural language and camera modalities. We have added a related discussion in the supplementary material of the main paper and will share our findings as soon as we obtain promising results.

***2. The generalizability of AlterMOMA on additional tasks.***
- Similarly, to demonstrate our method's generalizability across different tasks, we have conducted additional evaluations on an important task in autonomous driving, **multi-object tracking (MOT), as shown in Table 2 of supplementary PDF file.** We performed tracking-by-detection evaluation on the nuScenes validation dataset, and list the AMOTA of models pruned at 80% and 90% pruning ratios. The baseline model was trained with SwinT and VoxelNet backbones. Compared to the baseline pruning method ProsPr, AlterMOMA boosts the AMOTA of BEVFusion-mit by 1.9% and 3.1% for the two pruning ratios. These results further demonstrate our method's superior performance and generality across multiple tasks, including tracking.

***3. The running speed of models after pruning with AlterMOMA.***
- We agree that additionally reporting the running speed would make the paper more practical and convincing. Therefore, we report the running time in milliseconds. We tested the inference time of models after pruning using the structure pruning settings of the Table 5 of the main paper. Here are the performance results for BEVFusion-mit models with SECOND and ResNet101 backbones on an single RTX 3090: the inference times are **124.04 ms for unpruned models**, **106.39 ms for AlterMOMA-30%**, and **87.46 ms for AlterMOMA-50%**. These results highlight the improvements in inference speed achieved through our pruning techniques. The result table are also presented **in Table 3 of supplementary PDF file.**

***4. The computational overhead of proposed AlterMOMA.***
- The pruning process involves **one round** of *"Masking (LiDAR) - Reactivation (Camera) - AlterMasking (Camera) - Reactivation (LiDAR) - Importance evaluation"* before the beginning of each fine-tuning epochs, and the **total fine-tuning epoch number is 3**. In each 'Reactivation' stage, we only **sample 100 out of 62k batches** (the entire nuScenes training set comprises 62k batches with a batch size of 2). We tested the time for **the reactivation stage, which is around 170 seconds**. In the Importance evaluation stage, the important scores for parameters were evaluated by using the gradients via backpropagation before and after reactivation for 5 batches, as mentioned in Eqn.11-13 of the main paper. We tested the time for **the importance evaluation stage, which is around 10 seconds**. We tested the time for **the entire round, which is an average of around 190 seconds** in total on a single RTX3090 device. Subsequently, a 3 epochs fine-tuning is performed, which incurs the same computational cost as normal training.
Please note that unpruned baseline models, such as BEVFusion-mit, require a total of 30 epochs to train from scratch, including 20 epochs for LiDAR backbone pre-training and 10 epochs for the entire fusion architecture training. Since pruning is a one-time expense for generating a compact model, its overhead can be considered trivial, especially when considering the time-consuming nature of normal training. Due to the page limit, the aforementioned discussion will be included in the supplementary material for the camera-ready version.

***5. The features visualization.***
- We appreciate the insightful comment and acknowledge that adding feature visualizations of reactivated redundant features would make our paper more convincing. Therefore, we have included feature visualizations **in Figure 1 of the supplementary PDF file.** These figures will also be added to the camera-ready paper.

---

### Decision · Program_Chairs · 2024-09-25

**Decision:**

Accept (poster)

**Comment:**

he authors propose a novel pruning framework, AlterMOMA, which employs alternative masking to identify and prune redundant parameters in camera-LiDAR fusion models. The paper demonstrates the effectiveness of AlterMOMA through extensive experiments on the nuScenes and KITTI datasets, showing superior performance compared to existing pruning methods. The initial reviews consisted of two accepts and one borderline reject. The concerns raised were about the generalizability to additional modalities and tasks, as well as the running speed after pruning. In the rebuttal, the authors explained why camera-LiDAR is key for 3D object detection and demonstrated that the proposed method indeed speeds up the runtime after pruning. The final reviews resulted in two accepts and one weak accept. Therefore, the AC recommends acceptance.